# Characterizing immune variation and diagnostic indicators of preeclampsia by single-cell RNA sequencing and machine learning

Wenwen Zhou[1,2,10], Yixuan Chen [3,10], Yuhui Zheng [1,4,10], Yong Bai [1,10], Jianhua Yin[1,10], Xiao-Xia Wu[3], Mei Hong[2,5], Langchao Liang[4,6], Jing Zhang[3], Ya Gao[1], Ning Sun[3], Jiankang Li[1], Yiwei Zhang[3], Linlin Wu [7✉], Xin Jin [1,8,9✉] & Jianmin Niu [3✉]

Preeclampsia is a multifactorial and heterogeneous complication of pregnancy. Here, we utilize single-cell RNA sequencing to dissect the involvement of circulating immune cells in preeclampsia. Our findings reveal downregulation of immune response in lymphocyte subsets in preeclampsia, such as reduction in natural killer cells and cytotoxic genes expression, and expansion of regulatory T cells. But the activation of naïve T cell and monocyte subsets, as well as increased MHC-II-mediated pathway in antigen-presenting cells were still observed in preeclampsia. Notably, we identified key monocyte subsets in preeclampsia, with significantly increased expression of angiogenesis pathways and pro-inflammatory S100 family genes in VCAN+ monocytes and IFN+ non-classical monocytes. Furthermore, four cell-type-specific machine-learning models have been developed to identify potential diagnostic indicators of preeclampsia. Collectively, our study demonstrates transcriptomic alternations of circulating immune cells and identifies immune components that could be involved in pathophysiology of preeclampsia.

[1] BGI Research, Shenzhen 518103, China. [2] College of Life Sciences, South China Agricultural University, Guangzhou 510642, China. [3] Department of Obstetrics, Shenzhen Maternity & Child Healthcare Hospital, The First School of Clinical Medicine, Southern Medical University, Shenzhen 518028, China. [4] College of Life Sciences, University of Chinese Academy of Sciences, Beijing 100049, China. [5] Guangdong Provincial Key Laboratory of Protein Function and Regulation in Agricultural Organisms, South China Agricultural University, Guangzhou 510642, China. [6] BGI Research, Qingdao 266555, China. [7] Department of Obstetrics, the Eighth Affiliated Hospital, Sun Yat-sen University, Shenzhen 518033, China. [8] School of Medicine, South China University of Technology, Guangzhou 510006, China. [9] Shenzhen Key Laboratory of Transomics Biotechnologies, BGI-Shenzhen, Shenzhen 518083, China. [10]These authors contributed equally: Wenwen Zhou, Yixuan Chen, Yuhui Zheng, Yong Bai, Jianhua Yin. ✉email: lin.lin.wu@163.com; jinxin@genomics.cn; njianmin@163.com

Preeclampsia, a hypertensive disorder of pregnancy, with an incidence of about 2–5%, is one of the important causes of maternal and perinatal mortality[1]. The main clinical manifestations of preeclampsia are hypertension and proteinuria, with or without systemic organ dysfunction, and delivery of the fetal-placental unit is the only definitive intervention currently[2]. Therefore, more studies are needed to elucidate the pathophysiology of preeclampsia to improve treatment strategies and pregnancy outcomes.

As a syndrome with heterogeneous clinical manifestations and prognosis[3], the common pathological of preeclampsia include endothelial dysfunction, immune abnormalities, syncytiotrophoblast stress[4]. Unscheduled alterations of temporal events of inflammation (implantation), anti-inflammation (gestation), and inflammation (parturition) in normal pregnancy may lead to pathologic consequences, which is reported that dysregulated systemic immunity contribute to impaired angiogenesis and onset of preeclampsia[5]. Decidual immune cells are important contributors to the pathogenesis of preeclampsia, where researchers observed that innate immune cells such as monocytes and neutrophils were activated, CD4$^+$ T cells differentiation were imbalanced and B cells were stimulated by autoantigen[6]. Aberrant placental immunity may lead to changes in circulating immune cells. Several investigations reported the more abundance of IL-17-producing of peripheral lymphocytes[7], abnormal STAT5ab signaling in CD4$^+$ T cells and impaired regulatory T cells (Treg) signaling[8], together with the decreasing expression of Tim-3 pathway in CD8$^+$ T cells in patients with preeclampsia[9]. Such alternations may be associated with the imbalance of the maternal-fetal immune interface. Therefore, supplementing knowledge of immune cells in peripheral blood allows us to better understand the immune status of preeclampsia.

Transcriptome analysis is a powerful tool to study abnormal gene expression in diseases. Using RNA sequencing or microarray technology, previous studies have reported preeclampsia-associated genes in whole blood or peripheral blood mononuclear cells (PBMCs)[10–12]. By applying higher-resolution techniques, such as single-cell RNA sequencing (scRNA-seq), changes in gene expression can be pinpointed to a single cell, which can partially explain the heterogeneous immune response of diseases. For instance, two studies conducted scRNA-seq on preeclamptic placenta and determine the role of trophoblast cell subsets in placental immunity[13,14], while to the best of our knowledge, systematic changes of PBMCs in preeclampsia has not been well characterized by scRNA-seq.

In this study, we explored the transcriptomic characteristics of PBMCs in preeclampsia and normal pregnancies through scRNA-seq, revealing the abnormal expression of functional genes and pathways in circulating immune cells. Moreover, four cell-type-specific machine-learning models were constructed to distinguish preeclampsia from normal pregnancy, and provide insights of potential biomarkers for diagnosis at single-cell level. These works all deepen our understanding of preeclampsia.

## Results

### Clinical characteristics of subjects and profiling total PBMCs in preeclampsia and normal pregnancy.
In this study, we recruited 8 pregnant women diagnosed with preeclampsia (PE group, included one early-onset PE and seven late-onset PE) and 15 normal pregnant women (NP group) matched by propensity score matching to eliminate the effect of confounding factors (Table 1 and Supplementary Data 1). Blood samples were taken from individuals once they were diagnosed with preeclampsia. The gestational ages at sampling were all in third trimester, range from 31$^{+2}$ to 40$^{+3}$ weeks (Fig. 1a). All pregnant women enrolled

**Table 1 The clinical characteristics of PE and NP groups enrolled in this study, related to Fig. 1.**

| Factors | NP | PE | P-value |
|---|---|---|---|
| n | 15 | 8 | |
| Maternal age (years), median (IQR) | 29.0 (28.0, 31.0) | 29.0 (28.0, 31.5) | 0.87 |
| BMI (kg/m$^2$), mean (SD) | 25.5 (2.2) | 27.7 (1.8) | 0.03* |
| SBP (mmHg), mean (SD) | 114.0 (10.0) | 125.8 (20.2) | 0.07 |
| DBP (mmHg), mean (SD) | 68.1 (7.0) | 79.6 (13.7) | 0.01* |
| Cesarean Section | 1 (7%) | 4 (50%) | 0.03* |
| Preterm birth | 0 (0%) | 2 (25%) | 0.11 |
| Fetal gender | | | 1 |
| -Girl | 8 (53%) | 4 (50%) | |
| -Boy | 7 (47%) | 4 (50%) | |
| Fetal weight (g), mean (SD) | 3,221.3 (312.4) | 2,730.0 (682.1) | 0.03* |
| Gestational age at sampling (weeks), median (IQR) | 36.0 (31.9, 37.1) | 37.2 (36.2, 38.4) | 0.08 |
| Gestational age at delivery (weeks), median (IQR) | 39.6 (39.0, 40.7) | 37.3 (36.9, 38.7) | <0.01* |

Continuous variables were expressed as the mean with standard deviation (SD) and were tested for normality distribution with the Kolmogorov–Smirnov test. Independent t-tests were performed for normally distributed variables, and Mann–Whitney U-tests were performed otherwise. Categorical variables are presented as frequencies with percentages and were analyzed by the chi-square test or Fisher's exact test, as appropriate.
NP Normal Pregnancy, PE Preeclampsia, BMI Body Mass Index, SBP Systolic Blood Pressure, DBP Diastolic Blood Pressure, IQR Interquartile Range, SD Standard Deviation.

in the observation were singleton pregnancies and first pregnancies. The ages of pregnant subjects ranged from 24 to 34, and there was no bias of fetal gender and gestational age at sampling, while the gestation ages at delivery were significantly earlier in PE compared with the matched controls (37.3 vs. 39.6 weeks, $P < 0.01$), so were the fetal weights (2730.0 vs. 3221.3 g, $P = 0.03$). Subjects with preeclampsia had a significantly higher risk of cesarean section (50% vs. 7%, $P = 0.03$), higher diastolic blood pressure (79.6 vs. 68.1 mmHg, $P = 0.01$) and body mass index (27.7 vs. 25.5 kg/m$^2$, $P = 0.03$), some of which have been reported as the clinical presentation or risk factors (Table 1). Peripheral blood samples were collected from the 23 subjects and PBMCs were isolated, on which scRNA-seq was conducted and the transcriptomic features were then analyzed in the two groups (Fig. 1a). After data processing, we obtained 28,774 PBMCs from subjects in PE and NP (Supplementary Fig. 1a).

Dimensionality reduction clustering was performed on total PBMCs and projected onto 2D via UMAP. Based on the expression of canonical marker genes, we manually annotated five major cell types in the first clustering: T cells (CD3D$^+$CD3E$^+$), natural killer cells (NK, CD3D$^-$KLRD1$^+$), B cells (CD19$^+$CD79A$^+$), myeloid cells (AIF1$^+$CD14$^+$ or FCGR3A$^+$ or CD1C$^+$) and Platelet (PPBP$^+$PF4$^+$) (Fig. 1b–d, Supplementary Fig. 1b). The relative proportions of each major cell type have not changed in statistics (Fig. 1c), with exception of a significant decrease observed in total NK cells in PE (Fig. 1e). To further investigate the differences in cellular components and functions between PE and NP, we subclustered the T and NK cells, B cells and myeloid cells respectively and obtained 33 subsets of total PBMCs (Fig. 1b, c).

### Overall downregulation of B-cell-mediated immune responses and partial enhancement of MHC-II pathways in preeclampsia.
B cells mediate humoral immune response, and abnormal placental substance release to circulation may affect B cell activity.

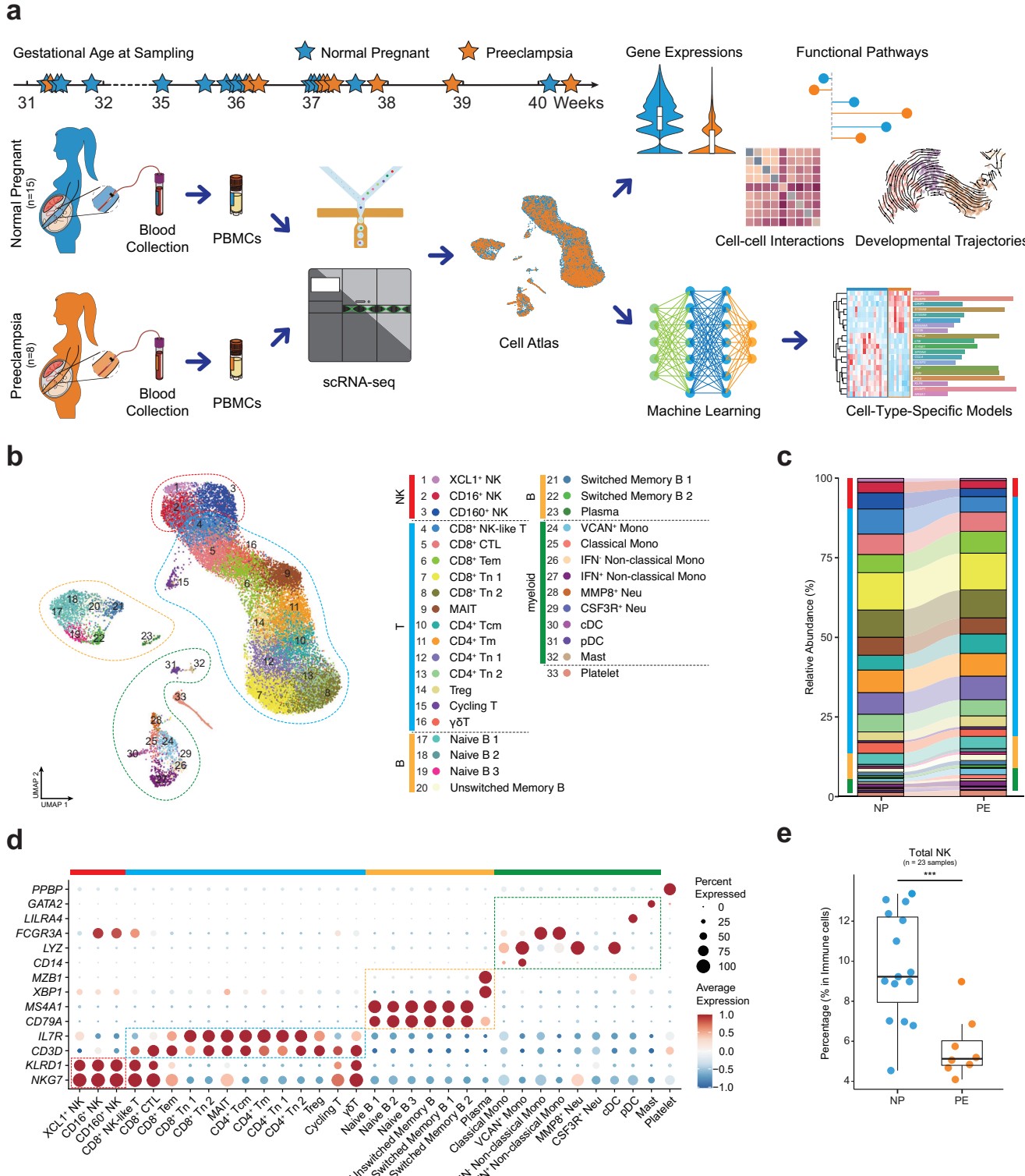

**Fig. 1 Study design and cellular composition of normal and preeclampsia pregnancy reveal by scRNA-seq. a** Schematic outline of the study design. Peripheral blood samples were collected from 8 preeclampsia women (PE) and 15 normal pregnant women (NP). **b** UMAP plot shows the overview of 33 clusters of PBMC subsets after clustering and subclustering and cell type annotation. Bars in different colors indicate the five major cell types. **c** Relative abundance of 33 subsets in NP and PE. Bars in different colors indicate the five major cell types. **d** Dot plot shows the expression of canonical markers which were used to identify the five major cell types. **e** Box plot shows the percentage of total NK cells in PBMCs of NP and PE, Student's *t*-test. ***P < 0.001. See also Table 1, Supplementary Fig. 1 and Supplementary Data 1–3.

According to the expression of *IGHD* and *CD27*, the B cells can be identified in two stages: naïve or memory[15]. We subclustered the total B cells and annotated three subsets of naïve B cells (Naïve B 1, Naïve B 2, and Naïve B 3, *IGHD⁺CD27⁻*), one subset of unswitched memory B cells (Unswitched Memory B, *IGHD⁺CD27⁺*), two subsets of switched memory B cells (Switched Memory B 1 and Switched Memory B 2, *IGHD⁻CD27⁺*), and Plasma cells (*XBP1⁺MZB1⁺*) (Fig. 2a, b). The relative

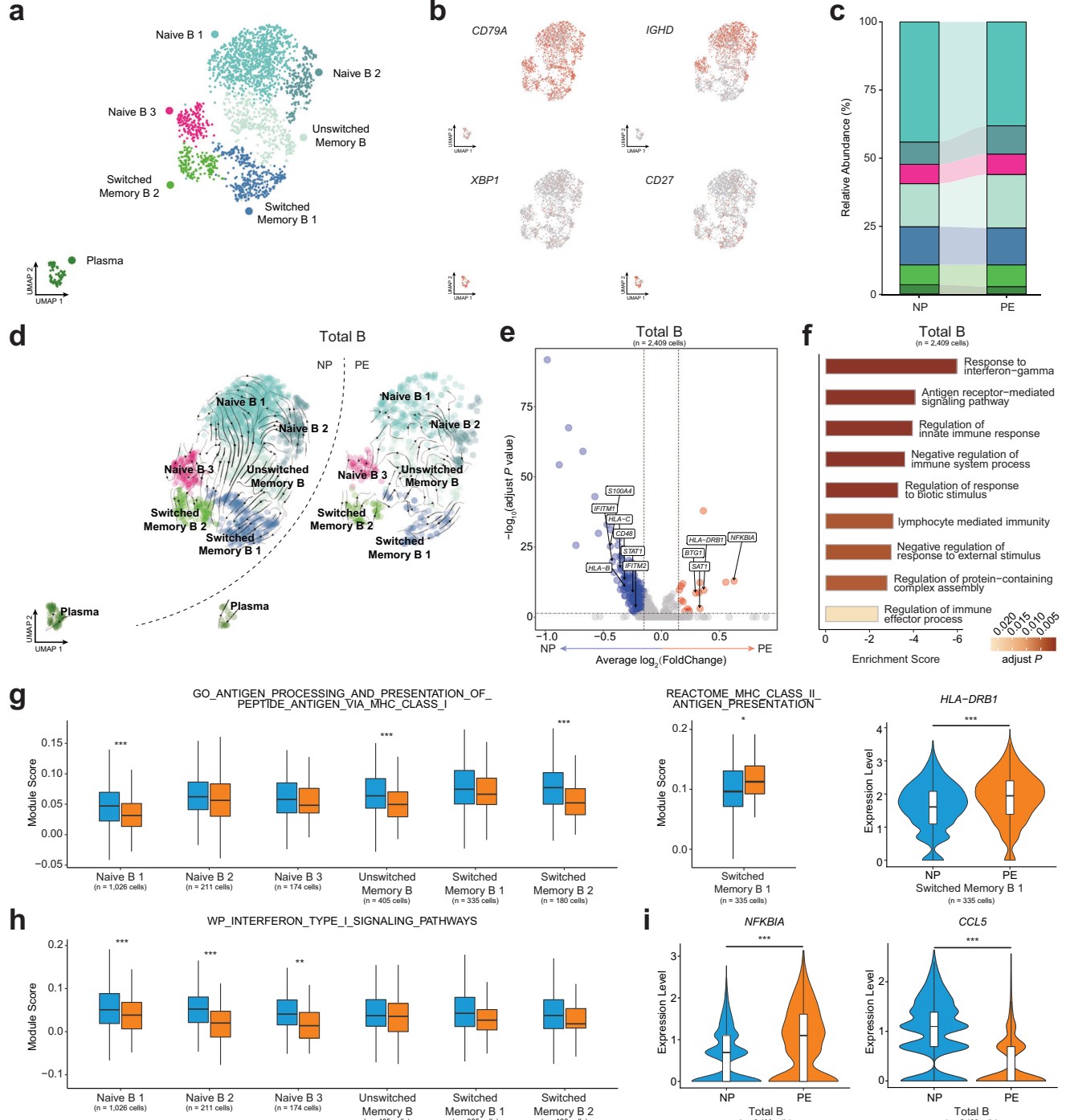

**Fig. 2 Sub-clustering and analyzing the function of B cell subsets. a** UMAP plot shows seven clusters of B cells after subclustering and annotation. **b** UMAP plots show the expression of canonical markers of B cell subsets in NP and PE. **c** Relative abundance of the seven clusters of B cell subsets in NP and PE. **d** UMAP plots show the developmental trajectories of B cell subsets in NP (left panel) and PE (right panel), revealed by RNA velocity analysis. **e** Volcano plot illustrates the significantly downregulated (blue dots) and significantly upregulated (red dots) DEGs in total B cells when comparing PE to NP, Wilcoxon rank-sum test. **f** Go enrichment analysis of downregulated DEGs of PE and bar plot shows the enrichment scores of immune-related pathways in total B cells. **g** Box plots show the expression of MHC-I and MHC-II-related pathways in B cell subsets, Student's *t*-test. Violin plot shows the expression of *HLA-DRB1* in Switched Memory B 1, Wilcoxon rank-sum test. *adjust *P* < 0.05, ***adjust *P* < 0.001. **h** Box plots show the expression of interferon type I signaling pathways in B cell subsets, Student's *t*-test. **adjust *P* < 0.01, ***adjust *P* < 0.001. **i** Violin plots show the expression of *NFKBIA* and *CCL5* in total B cells, Wilcoxon rank-sum test. ***adjust *P* < 0.001. See also Supplementary Fig. 2 and Supplementary Data 3.

proportions of B cell subsets have not changed in PE (Fig. 2c). RNA velocity was used to infer the developmental trajectories of B cell subsets, the differentiated direction from naïve B cells to unswitched memory B cells, and finally to switched memory B cells (Fig. 2d), was in line with our annotation.

We analyzed the differentially expressed genes (DEGs) in total B cells, and there were more downregulated DEGs in PE (Fig. 2e). We then performed enrichment analysis on these DEGs and uncovered the downregulation of immune processes in total B cells, including response to interferon-gamma and antigen receptor-mediated immune response (Fig. 2f). To further dissect the immune responses mediated by B cell subsets, gene set enrichment analysis was performed. Significantly decreased expression of antigen processing and presentation of peptide antigen via MHC-I pathway were observed in Naïve B 1, Unswitched Memory B and Switched Memory B 2, while the upregulation of MHC-II antigen presentation was discovered in Switched Memory B 1, together with the upregulation of MHC-II molecule *HLA-DRB1* in PE (Fig. 2g). The increased expression of MHC-II molecule *HLA-DRB1* was also observed in the placenta of patients with late-onset preeclampsia[16]. Moreover, interferon type I signaling pathway was significantly reduced in three naïve B cell subsets (Fig. 2h) and decreased expression of interferon-gamma response was observed in Naïve B 1 (Supplementary Fig. 2a), which were consistent with the GO enrichment results of overall downregulated immune response in total B cells (Fig. 2f). In addition, downregulation of platelet activation was observed in Switched Memory B 1 (Supplementary Fig. 2b) and the immunological memory formation process pathway was upregulated in Naïve B 3 (Supplementary Fig. 2c). Moreover, we observed that expression of *NFKBIA* and *BTG1* were upregulated and *CCL5* was downregulated in PE (Fig. 2i, Supplementary Fig. 2d). *NFKBIA* encodes the inhibitor of the NF-κB pathway, and *CCL5* is the targeted gene of NF-κB activity[17], which indicate the abnormal NF-κB signature in PE. *BTG1* was reported to suppress cell proliferative activity and maintain T cell quiescence[18], indicating the waning immune response mediated by B cells in PE, which were similar to the enrichment results of cell-free RNA (cfRNA) gene in the previous study[19].

Taken together, we observed an overall downregulation of immune response in total B cells and the enhancement of MHC-II pathways in Switched Memory B 1 of PE.

**Expansion of Treg and decreased expression of cytotoxic genes in T and NK cells of preeclampsia.** T and NK cells are the most abundant cell types in human PBMCs and our subclustering identified 16 subsets (Fig. 3a, b). According to the expression of canonical marker genes, we defined three NK cell subsets: XCL1+ NK cells (*NCAM1+XCL1+*), CD160+ NK cells (*CD160+FCGR3A+NCR1+*) and CD16+ NK cells (*CD160⁻FCGR3A+NCR1+*), and 13 T cell subsets: 4 subsets of naïve T cells (CD4+ Tn 1, CD4+ Tn 2, CD8+ Tn 1 and CD8+ Tn 2, *CCR7+SELL+*), 3 subsets of differentiated CD4+ T cells: CD4+ central memory T cells (CD4+ Tcm, *CCR7+CD44+*), CD4+ memory T cells (CD4+ Tm, *CCR7⁻CD44+*), CD4+ regulatory T cells(Treg, *FOXP3+IL2RA+*) and 4 subsets of differentiated CD8+ T cells: CD8+ mucosal-associated invariant T cells (MAIT, *SLC4A10+KLRB1+*), CD8+ cytotoxic lymphocytes (CD8+ CTL, *NKG7+GZMA+*), CD8+ effector memory T cells (CD8+ Tem, *CCR7⁻GZMK+*) and CD8+ NK-like cells (*CD3D+NCR1+*), one subset of cycling T cells (Cycling T, *CD3D+MKI67+*) and one subset of γδT cells (*CD3D+TRDV2+TRGV9+*) (Fig. 3c, Supplementary Fig. 3a).

The broad immunosuppression functions of Treg are reported, including suppressing immune cell proliferation and

cytotoxicity[20]. We observed the increasing Treg in PE, along with a significant decreased in CD8+ NK-like cells, CD160+ NK cells, and XCL1+ NK cells within the total T and NK cells (Fig. 3d). Given the cytotoxic function mediated by NK cells, we measured the expression of cytotoxic genes and revealed the lower expression levels of the cytotoxic genes in T and NK cells of PE (Fig. 3e). Similar to B cells, we observed the number of downregulated DEGs in total T and NK cells was more pronounced in PE (Fig. 3f). Compared with NP, there were lower expression levels of the lymphocyte activation pathway and interferon alpha beta signaling pathway in most T and NK cell subsets (Supplementary Fig. 3b). These dysfunctional states may link to the upregulation of suppressive *BTG1*[18] and the downregulation of some functional genes in total T and NK cells such as *CCL5* and *ITGAL*[21]. Additionally, the AP-1 transcription factor subunit genes (*FOS*, *FOSB* and *JUN*), which could modulate the activity of the immune system[22], were also downregulated in PE (Fig. 3f).

However, partial activations of T cell subsets were still identified. CD4+ Tn 2 was activated in PE, showing upregulation of unfolded protein response, response to interferon-alpha, oxidative phosphorylation, and DNA repair pathways (Fig. 3g). Increased expression of oxidative phosphorylation, DNA repair, regulation of T cell receptor signaling, and positive regulation of interferon-gamma production pathways were also observed CD8+ Tn 2 (Fig. 3h). Furthermore, *IFITM2* which may be correlated with pregnancy pathologies[23], and *GIMAP7*, one of the GIMAP family genes that are involved in apoptosis of peripheral lymphocytes and T helper cell differentiation[24], were upregulated in these two naïve T cell subsets (Fig. 3g, Supplementary Fig. 3c, d). In addition, *EST1* was reported to suppress the differentiation towards T helper 2 cells[25], upregulation of which in CD4+ Tn 2 may relate to the predominance of T helper 1 cells in preeclampsia[26]. And we observed increased expression of *IFI16* which plays an important role in preeclampsia[27], and another GIMAP family gene *GIMAP4*[24] were also identified in CD4+ Tn 2 of PE (Fig. 3g, Supplementary Fig. 3c). Similarly, the expression of *MYC*, *PDCD4* and *TMSB4X* reported to regulate the inflammatory response[28–30], were showed to increase in CD8+ Tn 2 (Supplementary Fig. 3d). All these functional DEGs and pathways suggested the preeclampsia-related activation of CD4+ Tn 2 and CD8+ Tn 2 in PE.

In addition, CD16-positive NK (CD16+ NK and CD160+ NK in our study) was in a more mature state than CD16-negative NK (XCL+ NK in our study)[31]. Then we investigated the change of the developmental trajectories of three NK subsets, which revealed a disparate differentiated direction of NP and PE (Fig. 3i), which suggests that the developmental trajectory of NK cells at disease state altered.

**Activation of VCAN+ Mono and IFN+ Non-classical Mono in preeclampsia.** Myeloid cells always function as antigen presenting cells (APCs) and mediated innate immune response and the monocytes were reported to relate to the inflammatory response of preeclampsia[32]. We identified four subsets of monocytes: VCAN+ monocytes (VCAN+ Mono, *VCAN+FCN1+*), classical monocytes (Classical Mono, *CD14+FCN1+*), IFN⁻ non-classical monocytes (IFN⁻ Non-classical Mono, *CD16+FCN1+IFNTM3⁻*) and IFN+ non-classical monocytes (IFN+ Non-classical Mono, *CD16+IFITM3+FCN1+*). Conventional dendritic cells (cDC, C1DC+), plasmacytoid dendritic cells (pDC, *ILIRA4+GZMB+*), Mast cells (*CPA3+KIT+*), MMP8+ neutrophils (MMP8+ Neu, *MMP8+S100A8+S100A9+*), CSF3R+ neutrophils (CSF3R+ Neu, *CSF3R+IRF1+*) were also annotated in our data (Fig. 4a–c, Supplementary Fig. 4a). The expansion of Classical Mono and

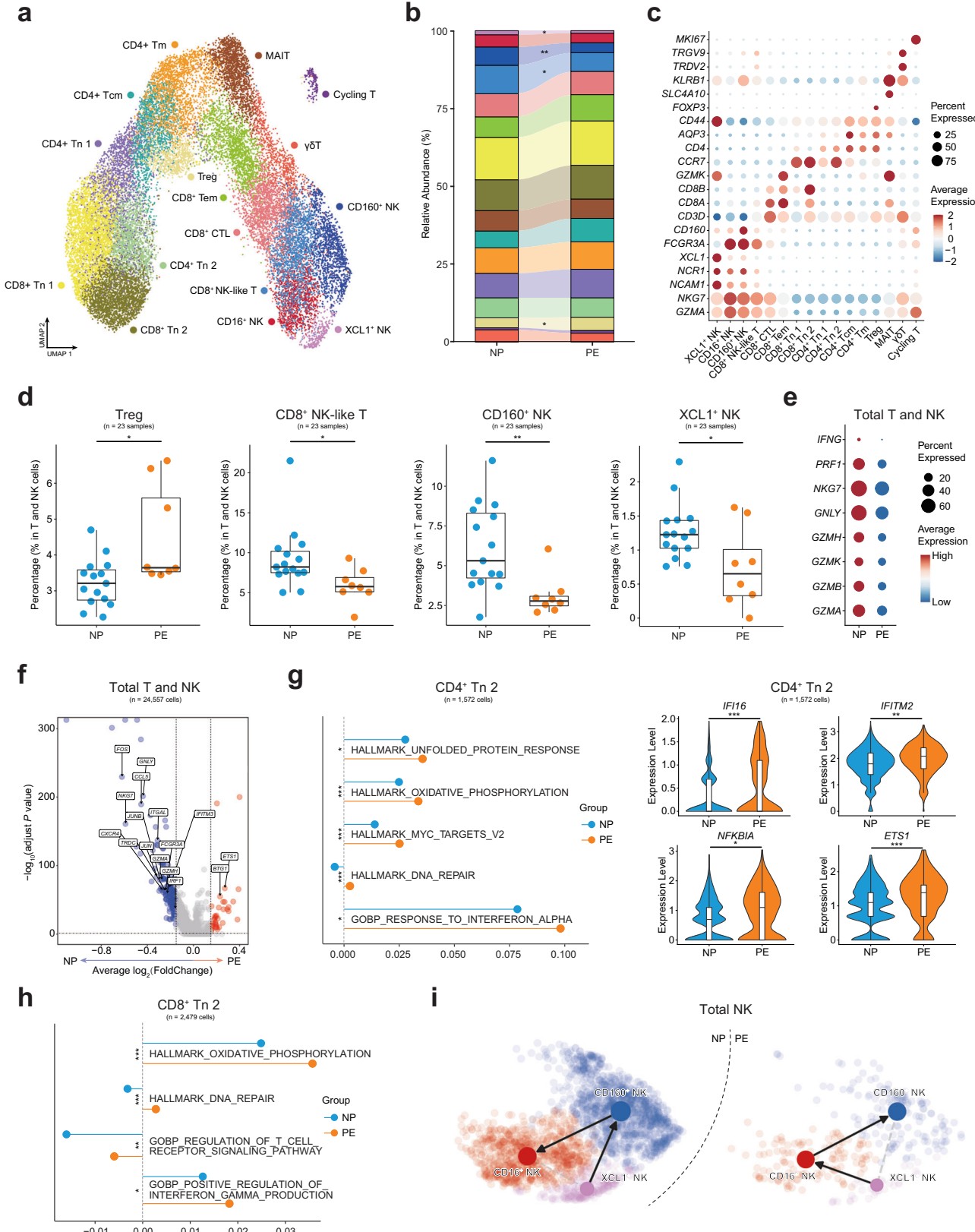

reduction of IFN⁻ Non-classical Mono were observed in PE (Fig. 4d), while the percentage of VCAN⁺ Mono and IFN⁺ Non-classical Mono of PE did not differ in NP (Fig. 4b), which suggests monocyte subsets playing distinctive roles in immune response of preeclampsia pregnancy. Additionally, the proportions of pDC and Mast cells also decreased in PE (Fig. 4d).

In line with the pathways analysis results of lymphocyte subpopulations, the downregulation of MHC-I and type I IFN response in several myeloid cell subsets were also identified in PE (Supplementary Fig. 4b). Both B cells and monocytes are important APCs. Consistent with the result in Switched Memory B 1, the MHC-II antigen presentation pathway was also

**Fig. 3 Subclustering and analyzing the function of T and NK cell subsets. a** UMAP plot shows 16 clusters of T and NK cell subsets of NP and PE.
**b** Relative abundance of the sixteen clusters of T and NK cell subsets in NP and PE. Significance of percentage change was evaluated with the Student's
*t*-test. *P < 0.05, **P < 0.01. **c** Dot plot shows the expression of canonical markers which were used to identify T and NK cell subsets. **d** Box plots show the
significantly changed percentages of T and NK cell subsets in total T and NK cells, Student *t*-test. *P < 0.05, **P < 0.01. **e** Dot plot shows the expression of
cytotoxic genes of total T and NK cells in NP and PE. **f** Volcano plot illustrates the significantly downregulated (blue dots) and significantly upregulated (red
dots) DEGs in total T and NK cells when comparing PE to NP, Wilcoxon rank-sum test. **g** Line and dot plot shows the expression of functional pathways in
CD4+ Tn 2, Student's *t*-test. Violin plots show the expression of functional genes in CD4+ Tn 2, Wilcoxon rank-sum test. *adjust *P < 0.05, **adjust
*P < 0.01, ***adjust *P < 0.001. **h** Line and dot plot shows the expression of functional pathways in CD8+ Tn 2, Wilcoxon rank-sum test. *adjust *P < 0.05,
**adjust *P < 0.01, ***adjust *P < 0.001. **i** UMAP plots show the developmental trajectories of three NK cell subsets in NP (left panel) and PE (right panel), the
arrows indicate the different directions. See also Supplementary Fig. 3 and Supplementary Data 3.

upregulated in three monocyte subsets (VCAN+ Mono, Classical Mono and IFN- Non-classical Mono) (Fig. 4e). In addition, the complement pathway is also upregulated in monocyte subsets (Fig. 4e), further indicating that monocyte-mediated immune responses were activated and contribute to the inflammatory response to preeclampsia[33].

Interestingly, we noted that VCAN+ Mono and IFN+ Non-classical Mono were activated with significantly higher expression levels of some key pathways, such as macrophage activation and interferon receptor activity in both two subsets (Fig. 4f, Supplementary Fig. 4c). Moreover, the upregulations of angiogenesis pathway in both two subsets and coagulation pathway in VCAN+ Mono were observed in PE (Fig. 4f, Supplementary Fig. 4c), these pathways were known to be of great importance to the occurrence and development of preeclampsia. Analysis of significantly upregulated and downregulated DEGs in VCAN+ Mono and IFN+ Non-classical Mono revealed some functional genes in PE, we found the decreased expression of chemokine *CCL5* and significant upregulation of *IFI30* and S100 family genes such as *S100A4, S100A6, S100A8* and *S100A10* in total myeloid cells of PE (Fig. 4g, h, Supplementary Fig. 4d, e). S100 family genes were reported to contribute to the inflammation of preeclampsia[34], and *S100A10* was one of the predictive gene of preeclampsia in previous study[10]. Additionally, there was no obvious difference in the developmental trajectories of the four monocyte subsets between the two groups (Fig. 4i).

Collectively, we uncovered monocyte subsets of VCAN+ Mono and IFN+ Non-classical Mono contributed to the inflammatory responses in PE and may be vital to the pathological process of preeclampsia.

**Imbalance of cellular interactions between circulating immune cell subsets in preeclampsia.** Dysregulation of the immune system is one of the characteristics of preeclampsia. To further explore alterations of cell-cell communication among circulating immune cell subsets in PE, we generate interaction networks among 33 PBMC subsets and found that the number of overall ligand–receptor pairs was reduced in PE, although the interactions of MMP8+ Neu with other cell types were significantly enhanced in PE (Fig. 5a), which may link to the activation of neutrophils in preeclampsia[35]. Then, we further analyzed the ligand-receptor pairs among these subsets in detail.

We first identified the significantly weakened cellular interactions in PE, which may suggest the lacking immune regulation in disease states. In NP, CD8+ Tem and CD8+ CTL were co-stimulated by APCs through CD28_CD86 and were co-inhibited through CTLA4_CD86 axis, CD8+ NK-like cells and CD160+ NK cells were suppressed through HLA-E_KLRC1 axis and activated through HLA-E_KLRC2 axis. Such stimulatory and inhibitory interactions of immune cell subsets were not pronounced in PE (Fig. 5b), which suggests that the imbalance of active and suppressive interactions may lead to an abnormal pregnant state. Then the stimulatory CD40_CD40LG,

TNFRSF13B_TNFSF13 and TNFRSF13B_TNFSF13B between B cell subsets with other immune cell subsets were weaker in PE (Fig. 5c). Moreover, CD4+ Tcm interacted with Classical Mono through CCL5_CCR1 axis in NP, CD16+ NK and CD8+ NK-like were interacted with Classical Mono through CCL3_CCR1 axis in NP but through CCL5_CCR1 axis in PE (Fig. 5d), which suggest different interactions preference. The platelet could be activated through SELP_SELPLG axis, the lower expression of this pair were observed in PE (Fig. 5d), which was consistent with the report of less activated platelet in preeclampsia[36]. Furthermore, almost all subsets stimulated pDC and cDC to produce pDC through FLT3_FLT3LG axis in NP, while little expression of this pair was observed in PE (Fig. 5d), which may be related to the decreased proportion of pDC in PE.

Of note, CD4+ T cell subsets were co-stimulated by APCs through CD28_CD80 and CD28_CD86 and co-inhibited by CTLA4_CD80 and CTLA4_CD86 in PE (Fig. 5e). Given that the higher affinity of CTLA4 to CD80/CD86, the inhibitory signals may be prominent in PE, which may explain the downregulation of the overall immune response of T cells. However, the inhibitory pairs (CTLA4_CD80/CD86) were not enhanced in CD4+Tn 2 and CD8+ Tn 2 with APCs (Fig. 5e), which may be associated with the partial activation of CD4+ Tn 2 and CD8+ Tn 2 in PE. In addition, lower expression of cytotoxic genes in PE should be associated with the intensification of TGFB1_TGFBR3, TGFB1_TGFbeta R1 and CD94:NKG2A_HLA-E (Fig. 5f). Moreover, enhancement of stimulatory pairs CD27_CD70, CD40LG_CD40 and TNFRSF13B_TNFSF13B were also observed in the interactions between Switched Memory B 1 and other four cell subsets (Fig. 5g), which may partially account for the upregulation of MHC-II pathway in Switched Memory B 1 of PE.

Overall, our results may explain the overall downregulated and partially upregulated immune responses in PE to some extent, emphasizing that the abnormal regulation and imbalance of stimulatory and inhibitory interactions may be associated with preeclampsia.

**Development of machine-learning models in the important subsets to diagnose preeclampsia.** Transcriptomic analysis of peripheral blood components uncovered a series of genes that could be used to diagnose or predict preeclampsia, or genes associated with preeclamptic pathophysiology. Several elegant studies have identified some predictive genes of cfRNA, which could predict the onset of preeclampsia before the clinical presentation[19,37,38]. As far as we know, there has been a lack of machine-learning models that specifically target circulating immune cells in patients with preeclampsia. Thus, we sought to develop cell-type-specific random forest (RF)-based classifiers for total Mono (VCAN+ Mono, Classical Mono, IFN- Non-classical Mono and IFN+ Non-classical Mono), CD4+ Tn 2, CD8+ Tn 2 and Treg to diagnose preeclampsia based on our scRNA-seq data and the results of the expression analysis above (Fig. 6a). Briefly, for each cell type, we first generated pseudo-cells from single cells

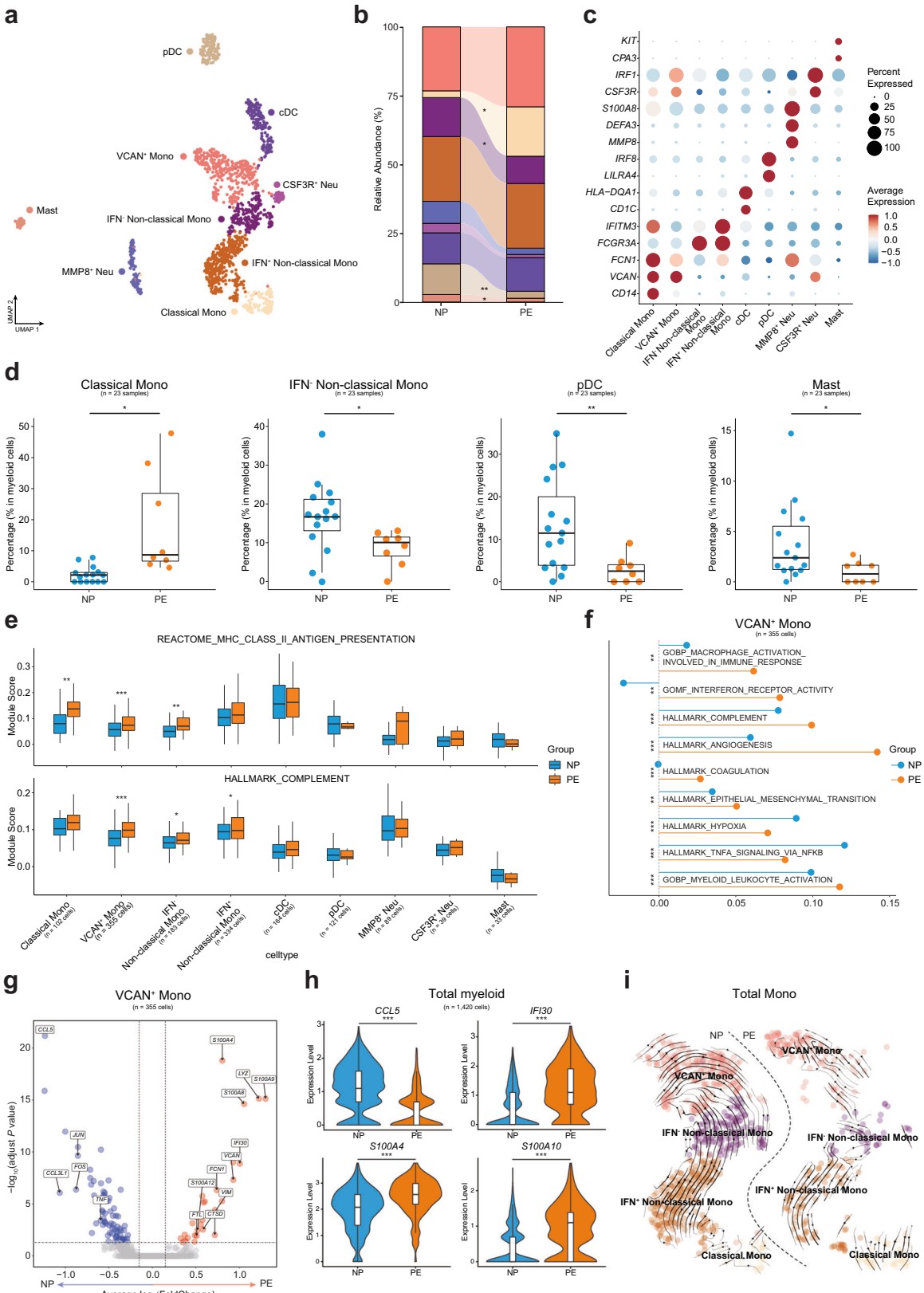

per individual after single-cell data processing, where each pseudo-cell was labeled as either positive or negative according to the status of the individual (PE or NP). The random stratified sampling was then applied on the pseudo-cell dataset to create a training set and an independent test set for ensuring the same ratio of positive and negative samples in the two sets

(Supplementary Table 1). Upon the training set after data normalization and feature selection, each cell-type-specific classifier was trained using the 5-fold cross-validation (CV) scheme (80% of random stratified samples used for training while 20% for internal validation). This procedure was repeated 100 times and the optimal hyperparameter values were determined based on the

**Fig. 4 Sub-clustering and analyzing the function of myeloid cell subsets. a** UMAP plot shows nine clusters of myeloid cell subsets of NP and PE. **b** Relative abundance of the nine clusters of myeloid cells in NP and PE. **c** Dot plot shows the expression of canonical markers which were used to identify myeloid cell subsets. **d** Box plots show the significantly changed percentages of myeloid cell subsets in total myeloid cells, Student's *t*-test. *adjust *P* < 0.05, **adjust *P* < 0.01. **e** Box plots show the expression level of functional pathways in all myeloid cell subsets, Student's *t*-test. **adjust *P* < 0.01, ***adjust *P* < 0.001. **f** Line and dot plot shows the expression of functional pathways in VCAN⁺ Mono, Student's *t*-test. **adjust *P* < 0.01, ***adjust *P* < 0.001. **g** Volcano plot illustrates the significantly downregulated (blue dots) and significantly upregulated (red dots) DEGs in VACN⁺ Mono when comparing PE to NP, Wilcoxon rank-sum test. **h** Violin plots show the expression of functional genes in total myeloid cells, Wilcoxon rank-sum test. ***adjust *P* < 0.001. **i** UMAP plots show the developmental trajectories of all monocyte subsets in NP (left panel) and PE (right panel), the arrows indicate the differentiate directions. See also Supplementary Fig. 4 and Supplementary Data 3.

highest AUROC (area under ROC curve) values (Fig. 6b, Supplementary Table 2). Due to the imbalanced data in this study (positive samples much less than negative ones), we additionally calculated weight F1 scores and confusion matrix to fairly quantify the classification performance. To do this, we utilized the Youden's J statistic to determine the optimal cutoff threshold for each classifier (Fig. 6b), leading to high values of the weight F1 scores (between 0.843 and 0.932) and confusion matrix results (Supplementary Fig. 5a, b). These results demonstrated an expressive performance of each cell-type-specific classifier for preeclampsia diagnosis during the training phase. The final cell-type-specific classifiers were built by re-training the existing ones with the optimal hyperparameters using the whole training sets.

Next, we used the independent test sets to evaluate the preeclampsia diagnosis capabilities of the four classifiers. To assess the stability of each classifier, we employed a bootstrapping approach by randomly sampling the same size of data from the independent test set with replacement 100 times. As a result, the large AUROC values of $0.986 \pm 0.011$ (mean $\pm$ SD) in total Mono, $0.893 \pm 0.040$ in CD4⁺ Tn 2, $0.988 \pm 0.007$ in CD8⁺ Tn 2, and $0.867 \pm 0.051$ in Treg were reached, respectively (Fig. 6c). Using the optimal cutoff threshold, we additionally calculated the sensitivity (SEN, mean value ranging from 0.821 to 0.936), specificity (SPE, from 0.828 to 1.000), negative predictive value (NPV, from 0.927 to 0.981), and positive predictive value (PPV, from 0.587 to 1.000) of these four models (Fig. 6c), as well as the weighted F1 scores (from 0.839 to 0.942) and confusion matrix results for these four classifiers (Supplementary Fig. 5c, d). Collectively, these results suggest that our models are well-performing to distinguish preeclampsia in both training set and independent test set.

The four sets of gene features that were identified by our feature selection method effectively discriminated between NP and PE. Some of the gene features were commonly identified across all four classifiers, while others were specifically found in total Mono, CD4⁺ Tn 2 cells, CD8⁺ Tn 2 cells, or Treg (Fig. 6d), which implied the necessity for developing models at the subset level. We prioritized and ranked the gene features in each of the four sets according to the mean absolute SHapley Additive exPlanation (SHAP[39]) values that were computed across all samples in the corresponding independent test set. A SHAP value (also called feature importance) represented the contribution of a gene feature towards distinguishing PE from NP. We then evaluated the performance of top 20 gene features and visualized them for each of the four classifiers. After normalization, we noted that the 23 samples were accurately divided into NP and PE groups (Fig. 6e–h). Notably, *DUSP1* and *FOS* were identified by all classifiers with high feature importance, while *NFKBIA* was selected by three classifiers that were associated with T cell subsets (Fig. 6e–h). The protein level of *DUSP1* was reported to decrease in severe preeclampsia and was consistent with our results[40]. Additionally, *FOS* was downregulated in PE and *NFKBIA* was upregulated in most immune cell subsets in above functional analysis results. Furthermore, in the classifier of Treg, we

observed a decreased expression level of *SPON2*, *NFKBIA*, and the MHC-I molecule *HLA-C* in PE (Fig. 6e), which were in line with the downregulation of immune response in PE. *SPON2*, encoding extracellular matrix protein mindin, is essential for the recruitment activities of inflammatory cells as an integrin ligand[41]. Similarly, *GIMAP7*, which was one of the upregulated DEGs in CD4⁺ Tn 2, was also selected in the classifier of CD4⁺ Tn 2 (Fig. 6f). Cytotoxic genes such as *GNLY*, *NKG7*, *GZMA*, *GZMH* were selected in the classifier of CD8⁺ Tn 2 (Fig. 6g), which were tied well with the lower expression of these genes in total T and NK cells of PE (Fig. 3e). In the classifier of total Mono, the functional genes of monocytes including S100 family genes *S100A8* and *S100A9*, monocyte-specific markers *LYZ*, *MS4A6A* which reported to regulate during myelomonocytic differentiation[42] and *CD36* which involved in angiogenesis and inflammation[43] were also identified (Fig. 6h). These results indicated that our machine-learning models, after feature selection and model training, have found similar discoveries to the functional analysis results and have also determined genes that may be related to disease pathophysiology.

## Discussion

The immunological status of pregnancy is dynamic and heterogeneous[44]. Disruption or dysregulation of immune mechanisms may lead to pregnancy complications[45], including preeclampsia[46]. Using scRNA-seq, we profiled 28,774 PBMCs from 8 preeclampsia patients and 15 matched controls to investigate immunological variations in preeclampsia in the third trimester. Our findings fit well with the accepted knowledge that specific monocyte subsets as potential contributors to the inflammatory response in preeclampsia. However, we also noticed that the immune cells in preeclampsia did not be activated all the time, as we observed downregulation of immune activities in several PBMC subsets.

Previous studies have reported a descended proportion of Treg cells in both peripheral blood and decidual immune cells of preeclampsia[8,47,48]. However, the use of different technologies, definitions[49], and gestational ages may contribute to the diverse changes of Treg cells. For instance, recent research has reported the expansion of decidual Treg cells in preeclampsia patients[50]. Uterine Treg with inflammatory characteristics could be amplified by triggers in pregnant mice which then led to pregnancy demise or preeclampsia-like features[51], which were in contrast with prior observations. In our investigation, we found an elevated proportion of Treg in total T and NK cells of preeclampsia, then identified the corresponding alternations of suppressive immune response:

Firstly, the downregulation of chemokine *CCL5* in most immune cell subsets indicates the migration activity mediated by *CCL5* may be impaired in preeclampsia[17], together with the upregulation of *BTG1* and *NFKBIA*, which have suppressive functions or can inhibit the NK-*κ*B pathways, respectively. In PE, we observed downregulation of lymphocyte activation and lymphocyte-mediated immunity pathways, which were also

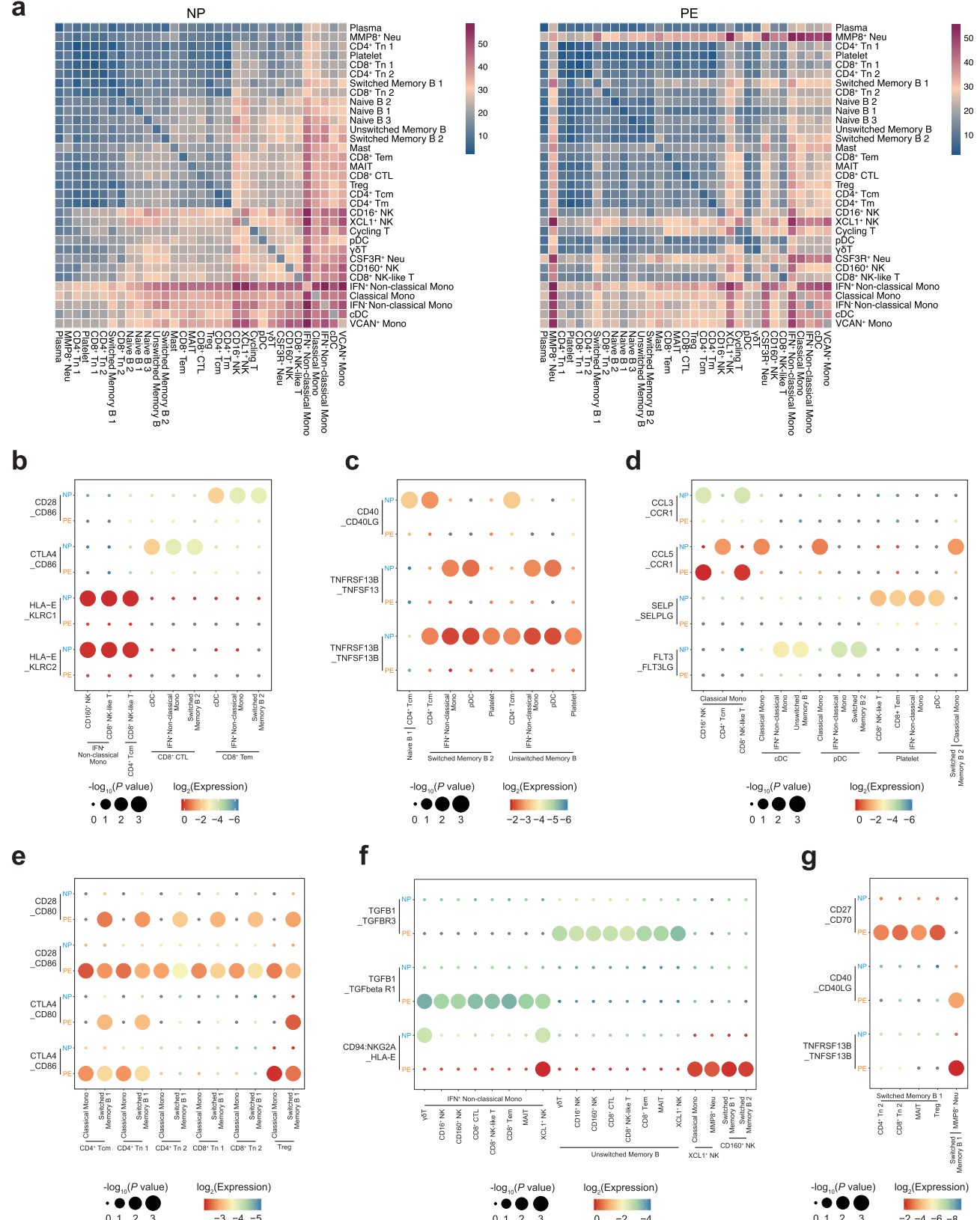

uncovered in the gene set enrichment analysis of circulating cfRNA and peripheral blood, including the T-cell receptor complex, B-cell receptor signaling pathway, and humoral immune response[19,52]. In addition, we observed a reduction in the expression of antigen processing and presentation via MHC-I and interferon type I signaling pathway in preeclampsia like prior reports[19,52]. Above all, the downregulation of circulating immune responses may be a sign of preeclampsia pregnancy. Secondly, the proportion of NK cells in total T and NK cells together with expression of cytotoxic genes decreased in PE. Preceding studies demonstrated diverse changes in proportions of NK cell subsets in preeclampsia, such as the reduction of vascular endothelial

**Fig. 5 Cellular interaction of immune cell subsets in normal pregnancy and preeclampsia. a** Heatmaps show the interaction counts between 33 immune cell subsets in NP (left panel) and PE (right panel). **b** Dot plot shows the interactions between cytotoxic cell subsets and other cell subsets through the expression of stimulatory and inhibitory pairs in NP and PE. **c** Dot plot shows the interactions between B cell subsets with other cell subsets through the expression of stimulatory pairs in NP and PE. **d** Dot plot shows the interactions among immune cell subsets through the expression of chemokine-receptor pairs and other functional pairs in NP and PE. **e** Dot plot shows the interactions between T cell subsets and APCs through the expressions of co-stimulatory and co-inhibitory pairs in NP and PE. **f** Dot plot shows the interactions between cytotoxic cell subsets and APCs through the expression of inhibitory pairs in NP and PE. **g** Dot plot shows the interactions between Switched Memory B 1 and other cell subsets through the expression of stimulatory pairs in NP and PE. See also Supplementary Data 3.

growth factor-expressing NK cells[53] and significantly decreased percentage of CD56[+]/NKp46[+] cells and CD56[bright]/NKp46[+] cells in peripheral blood of PE[54], while the expansion of CD69[+]/CD56[Dim]/CD16[+] cytotoxic NK cells was observed at delivery[55], which indicate that the changes of NK cells varied from different cell subtypes and gestational ages. Decreased expression of cytotoxic genes was also discovered in a subset of trophoblast cells[13]. And Sargent et al. have suggested that inadequate stimulation of decidual NK cells, which are involved in the synthesis of immunoregulatory cytokines and angiogenic factors, may lead to poor trophoblast invasion[56]. Similarly, it was suggested that the reduced cytotoxicity of decidual NK in PE may be due to the expansion of Treg and elevated expression of TGFb1 in decidual[50], which were in line with our findings in the peripheral blood of PE.

Despite the downregulation of the immune response in PE, we still found essential activated subsets, such as several APCs (Switched Memory B 1, Classical Mono, VCAN[+] Mono and IFN[-] Non-classical Mono) with upregulation MHC-II-mediated antigen presentation pathways, indicating that APCs were still exposed to exogenous antigens and stimulated. Moreover, partial activations were noted in two naïve T cell subsets, represented by the upregulation of some hallmark pathways such as oxidative phosphorylation and IFN-related pathways.

More importantly, monocytes were reconstructed and activated in PE, particularly the IFN[+] Non-classical Mono and VCAN[+] Mono. The upregulation of immune pathways (such as complement, myeloid leukocyte activation, interferon receptor activity, and macrophage activation) was observed in these monocyte subsets, which were aligned with the known systematic inflammation of preeclampsia. Additionally, the upregulation of coagulation and angiogenesis process may correlate with reported preeclampsia pathologies of preeclampsia development. Finally, the pro-inflammatory S100 family genes were found to be upregulated these monocyte subsets, which could play roles in preeclampsia by contributing to inflammation or interacting with other factors in preeclampsia[34].

A handful of studies has built machine-learning models to classify or diagnose disease or pregnancy status using scRNA-seq data[57–59]. Here, we utilized circulating immune cells in pregnancy to construct cell-type-specific models for the diagnosis of preeclampsia. Apart from the accurate and effective discrimination of preeclampsia from normal pregnancy at the single-cell subpopulation level, these models also identified the specific genes that may contribute to the disease's pathological processes. Although the lack of scRNA-seq data from PBMCs of preeclampsia prevented us from further validating the performance of our models in external datasets, the successful try of developing the specific models that accounted for cell types heterogeneity may enhance the accuracy of preeclampsia diagnosis. We also provide a framework for analyzing disease characteristics. We could identify disease-related transcriptomic changes at one time and construct cell-type-specific models for key subsets of interest, which may be used as potential biomarkers for diagnosis or therapeutic targets to promote understanding of disease pathophysiology and clinical applications.

In conclusion, on the one hand, we identified activated APCs in PE and inflammation contributors in preeclampsia, which were consistent with the previous understanding of the enhanced inflammatory response in preeclampsia. On the other hand, the aberrant expansion of Treg and reduction of cytotoxic genes expression, suggests that protective immunity was not activated effectively. Together demonstrated the dysregulation of immunity in preeclampsia.

Our study has some limitations, primarily due to difficulty in sample collection. As a cross-sectional study, this study lacks a longitudinal sampling and comparative study between early- and late-onset preeclampsia, mild and severe preeclampsia. Furthermore, the expansion and reduction of immune cell subsets were not validated by in vitro experiments. Finally, the limited sample size involved in the current study may introduce bias to our models for preeclampsia diagnosis, and therefore, further investigations were needed to validate the performance using additional scRNA-seq data. However, the results of AUROC, weight F1 scores and other evaluation metrics obtained from the training sets and independent test sets showed the reliability of our classification models, which can also be demonstrated by the identified preeclampsia-associated gene features that were consistent with previous studies.

## Methods

**Sample collection**. This study is based on the birth cohort in Shenzhen. The cohort was collected to assess the long-term cardiovascular risk of mothers and offspring exposed during pregnancy. Women with singleton pregnancies were recruited at 6 to 8 weeks of gestational age in Shenzhen Maternity and Child Health Care Hospital. Regular antenatal examination was performed. Blood samples were taken from individuals once they were diagnosed with preeclampsia, and the pregnancy outcomes were recorded. Preeclampsia were diagnosed according to the Guidelines for the Diagnosis and Treatment of hypertensive Disorders in Pregnancy (2020) of the Chinese Society of Obstetrics and Gynecology, Chinese Medical Association[60].

We then performed one-to-two propensity score matching to match uncomplicated cohort participants as controls based on maternal age and gestational age of blood sampling. Some of the hemolysis samples were eliminated. Finally, a total of 8 individuals with preeclampsia and 15 normal pregnant women were included in this study, all of whom were women aged from 20 to 34 years old and who were singleton pregnancies and first pregnancies.

**Ethics statement**. All pregnant women enrolled in the study were required to fill in the birth cohort baseline questionnaire and sign the Shenzhen Birth Cohort informed consent form. This study was approved by the Ethics Committee of Shenzhen Maternity and Child Health Care Hospital (Shenzhen Maternal and Child

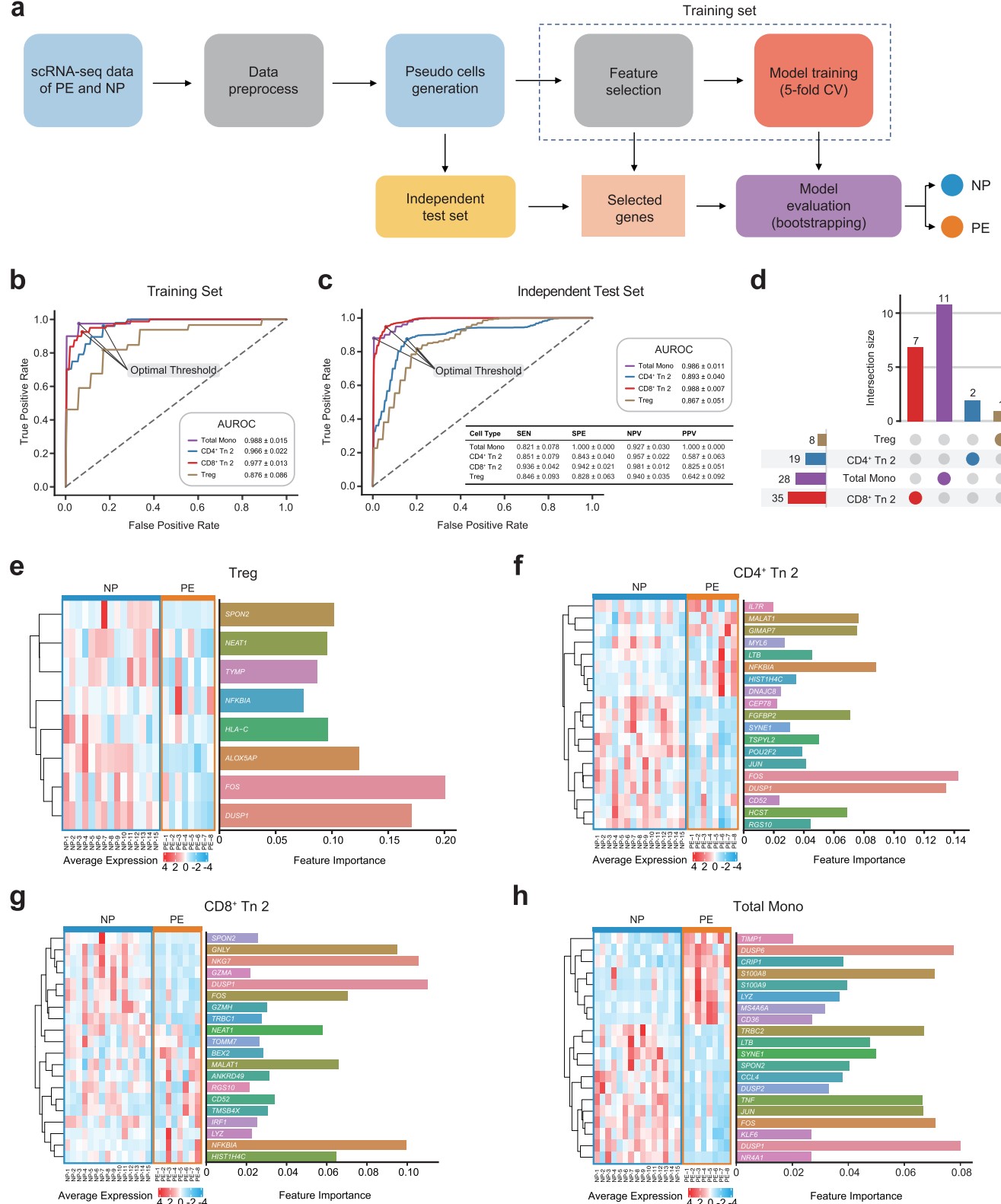

**Fig. 6 Develop machine-learning models to diagnose preeclampsia and evaluate the model performance. a** Flowchart of developing cell-type-specific machine-learning models based on the scRNA-seq data of PBMCs from PE and NP. **b**, **c** The results of evaluation metrics showing the performance of the four cell-type-specific models for preeclampsia diagnosis using the training sets (**b**) and the independent test sets (**c**). AUROC: Area Under ROC Curve, SEN: Sensitivity, SPE: Specificity, NPV: Negative Predictive Value, PPV: Positive Predictive Value. **d** Upset plot shows the number of features in the four models respectively, the horizontal columns indicate the number of features in each model and the vertical columns indicate the number of cell-type-specific features in each model. **e-h** Top 20 important features determined and ranked by SHAP method, the horizontal columns indicate feature importance (right panel) and heatmaps show the average expression of selected gene features in individuals in each model, including Treg (**e**) CD4+ Tn 2 (**f**) CD8+ Tn 2 (**g**) and total Mono (**h**). See also Supplementary Fig. 5, Supplementary Table 1, 2 and Supplementary Data 3.

Ethics Review SFYLS [2021]031). All ethical regulations relevant to human research participants were followed.

**Collection of peripheral blood samples and isolation of PBMCs**. 3 mL of peripheral venous blood was collected using a blood collection tube with EDTAK2. The blood was then mixed with 3 mL of sterile phosphate-buffered saline (PBS, Cat. No. 10010–031) and Histopaque-1077 (Cat. No.10771) and centrifuged at 500 g for 20 min. The middle layer of mononuclear cells was transferred into a new 15-mL conical centrifuge tube. The middle layer of mononuclear cells was transferred to a new tube and washed twice with PBS containing 1% bovine serum albumin (BSA). The cells were resuspended in 2 mL of 1% BSA (PBS) and cell viability was checked using Trypan blue (Cat. No. C0040) staining. About 100,000 cells were extracted and centrifuged at 300 g for 5 min, then resuspended with 100 μL of Cell Resuspension Buffer (Cat. No. 1000019895). The sample preparation works were conducted at room temperature.

**scRNA-seq library preparation and sequencing with DNBelab C4 system**. DNBelab C4 system (BGI, v1.0) was used for library preparation and sequencing[61]. In brief, the single-cell suspension of PBMCs, together with functionalized beads and lysis buffer were input into a pressure-driven microfluidic device and emulsion droplets were generated. The single cell in the droplet was lysed and mRNA transcripts were released, which were later captured by barcoded beads. Then the emulsion was broken and filtered, mRNA is reverse transcribed into cDNA and then cDNA was amplified and sheared to shorter fragments. After adapter ligation and cyclization to single-strand circular DNA of making sequencing library, the later amplified DNA nano balls (DNB) were patterned on nanoarrays and sequenced on the BGISEQ-500 sequencer.

**scRNA-seq data processing**. Seurat object was created by the Seurat R package (v4.3.0)[62], with R (v4.1.1)[63]. The genes expressed in >3 cells, and the cells with the number of genes detected ranged from 200 to 6000, with <10% of the mitochondrial genome and <1% of 10 hemoglobin genes (*HBA1, HBA2, HBB, HBD, HBE1, HBG1, HBG2, HBM, HBQ1, HBZ*) were retained. For quality control, the ribosomal genes, mitochondrial genes, pseudogene genes, and long intergenic non-coding RNAs (lincRNAs) were excluded from further analysis.

**Dimension reduction and batch effect correction**. To eliminate the influence of the batch effect, we scaled and transformed data, and identified variable genes using the "SCTransform" function and applied the "RunPCA" function to perform linear dimensionality reduction. Then the PCA matrix was fed into the "RunHarmony" function of harmony R package (v0.1.0)[64] for batch correction. We used the batch-correct matrix for clustering by the "FindNeighbors" and "FindClusters" functions, the Louvain algorithm. Finally, we used the first 19 principal components (PCs) for UMAP dimensionality reduction by the "RunUMAP" function (n.neighbors = 30 L, min.dist = 0.4) to map the clustering results to 2D.

**Major cell type annotation**. The first round of clustering of all cells using 0.3 resolution, while the "FindAllMarkers" function was used to find marker genes for each cluster, identified six major cell types, including T cells (*CD3D, IL7R*), NK cells (*NKG7, KLRD1*), B cells (*CD79A, MS4A1*) and Plasma cells (*XBP1, MZB1*), myeloid cells (*LYZ, FCGR3A*) and Platelet (*PPBP*).

**Subclustering and cell type annotation**. To identify cell subsets with more granularity in the major cell types, we performed a second round of clustering on T & NK cells, B & Plasma cells and myeloid cells. The steps of the second round of clustering are basically the same as those of the first round, with the resolution of 1.2, 0.5 and 0.3 respectively. Single cells expressing classical marker genes for both major cell types were labeled as doublet and were excluded from downstream analysis. Overall, the five major cell types were further identified into 33 subsets, included three subsets of NK cells, thirteen subsets of T cells, seven subsets of B cells, nine subsets of myeloid cells and one subset of Platelet.

**Differential expressed genes (DEGs) analysis and GO enrichment analysis**. To identify DEGs between two groups, we used the "FindMarkers" function in the Seurat R package. Genes with adjust *P*-values < 0.05 and absolute values of average $\log_2$(Fold-Change) > 0.15 were labeled as significant DEGs. The "enrichGO" function (OrgDb = org.Hs.eg.db, pvalueCutoff = 0.05, qvalueCutoff = 0.05, pAdjustMethod = "BH") of clusterProfiler R package (v4.2.2)[65] was used to perform functional analysis of significantly up- and downregulated genes respectively.

**Cell module scores evaluating**. We evaluated the module scores by applying Hallmark and custom gene sets to each cell using the "AddModuleScore" function of the Seurat R package. These gene sets were collected from the Molecular Signatures Database (MSigDB)[66].

**Cell-cell interaction analysis**. We used CellPhoneDB (v3.1.0)[67] to analyze cell-cell interactions between immune cells. The cell-type specificity of receptor-ligand complexes was calculated from the default database, and potential receptor-ligand interaction networks were derived by ranking and screening highly specific interactions between cell types according to the number of significant receptor-ligand pairs enriched between the two cell subsets.

**RNA velocity estimation**. The spliced and unspliced matrices were obtained first by recalculating the previous aligned bam files of scRNA-seq data using "Velocyto" (v0.17.16)[68]. The spliced and unspliced matrices of the associated cell annotation information were then basically preprocessed using the "scVelo" Python package (v0.2.4)[69], and the "scvelo.tl.recover_dynamics" and "scvelo.tl.velocity" (mode = "dynamical") function were used to estimate the RNA velocity. The "scvelo.tl.paga" function was also used to infer trajectory relationships.

**Data preprocessing and splitting**. We first filtered out unwanted genes and single cells, and then normalized the expression data over 10,000 for each single cell, followed by performing logarithm transformation and selecting 2,000 highly variable genes (HVGs). Due to the large number of single cells a patient possessed as well as the inherence of sparsity and uneven RNA capture in single-cell, it is undesirable to build a model to diagnose PE based upon such dataset. Hence, for each annotated cell type, we computed the mean gene expression of every 5 single cells that were randomly sampled (without replacement) from the same patient, denoted as pseudo-cells. Towards each cell-type-specific pseudo-cell dataset, a training set and an independent test set were constructed with the exertion of the stratified random splitting approach in a 7:3 ratio. We further removed the mean value of the expression from each gene and scaled to unit variance in the training data. The mean value and variation of each gene were preserved and then applied to the corresponding gene in the independent test dataset.

**Feature selection**. To eliminate redundant genes, we employed two feature selection methods, the mutual information (MI)-based algorithm and the Boruta algorithm[70]. The MI-based technique is one of the filter methods that calculate feature weights based on mutual information by considering the relationship between the features and class labels. The Boruta algorithm is a widely used wrapper method to select subset features with their importance measurements greater than the highest feature importance obtained by permuting a copy of features across samples to destroy the relationship between features and class labels. Genes were only included in the diagnosis model if they were identified by both algorithms in the training dataset. The selected genes were then applied to the independent test dataset to filter out any irrelevant genes.

**Machine learning model training and evaluation**. Using each cell-type-specific training dataset, the random forest (RF) classification model implemented in scikit-learn (v1.0.2) was trained with considering class weight due to the imbalanced dataset. The optimal hyperparameters of the RF model were determined by maximizing AUROC using the random search strategy in a 5-fold cross validation. The independent testing dataset was then used to evaluate the performance of the classification model.

Moreover, we applied the SHAP method (version 0.39.0)[39] to determine the importance of genes in predicting PE. Each SHAP value measured the change in the predicted value of PE for patient $i$ attributed to gene $j$. Mean absolute SHAP values across all patients in the dataset represented the overall importance of a particular gene in the diagnosis of PE by the RF model. A larger mean absolute SHAP value of a gene represented a higher contribution towards PE prediction. Genes were initially ranked by the mean absolute SHAP values that were calculated from the training dataset and we selected the top 20 genes to investigate their contributions towards PE.

**Statistics and reproducibility**. We conducted a single-cell RNA sequencing study using a cohort of 23 samples, which were segregated into two groups based on clinical diagnosis: 8 preeclampsia cases and 15 normal pregnancy cases. None of the same sample was measured repeatedly. Maternal age, gestational age at blood collection, and infant gender at birth did not exhibit any significant differences between the two groups. Statistical analyses were conducted using R software. Boxplots and dot plots were utilized to visualize the distribution of data for cell proportions and pathway expression, and $P$-values were calculated using Student's $t$-test. To compare gene expression levels, we employed violin plots and volcano plots for data visualization, and the Wilcoxon test was used to compute $P$-values, with the Bonferroni correction applied to obtain adjusted $p$-values. In the figures, $P$-values and adjusted $P$-values >0.05 were considered to be statistically non-significant and were not labeled, whereas values less than or equal to 0.05, 0.01, and 0.001 were annotated with *, **, and ***, respectively. The exact $P$-values are provided in Supplementary Data 3. The statistical tests employed are referred to in the respective figure legends.

**Reporting summary**. Further information on research design is available in the Nature Portfolio Reporting Summary linked to this article.

## Data availability

The normalized gene expression matrix of scRNA-seq data that supported the results was deposited in the CNGB Nucleotide Sequence Archive (CNSA)[71] (accession code: CNP0003201; https://db.cngb.org/search/project/CNP0003201/) of the China National GeneBank DataBase (CNGBdb)[72]. Source data underlying Figueres are in Supplementary Data 3. Other relevant data are available from the corresponding author upon reasonable request.

## Code availability

Details of publicly available software used in the study are given in the Methods. Custom codes of machine learning are available at https://github.com/y-bai/pbmc-pe/tree/v1.0.0 and https://doi.org/10.5281/zenodo.10223665[73].

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

## Acknowledgements

This work was supported by the National Natural Science Foundation of China (81830041, 32000398, 32171441, 72374227), Shenzhen Science and Technology Program (JCYJ20220818103608017, JCYJ20220818103607015, JSGG20210802152800001), and National Key R&D Program of China (2022YFC2502400). We also would like to thank all subjects who contributed the sample and China National GeneBank who contributed computational resources and scRNA-seq support for this work.

## Author contributions

J.M.N., X.J., and J.H.Y. designed and supervised the study. L.L.W. and Y.X.C. collected clinical samples and clinical information. X.X.W., J.Z., Y.W.Z., and N.S. performed the experiments. Y.H.Z., W.W.Z., L.C.L., and J.K.L. analyzed the data. Y.B. developed the machine learning models. W.W.Z., Y.H.Z., Y.X.C., J.H.Y., and Y.B. wrote the manuscript. W.W.Z., J.H.Y., X.J., J.M.N., Y.G., and M.H. participated in the manuscript editing and discussion.

## Competing interests

The authors declare no competing interests.
