## [Peer Review File · Communications Biology]

Reviewers' comments:

Reviewer #1 (Remarks to the Author):

Thank you for inviting me to review the article -- Dissecting circulating immune cells of preeclampsia by single-cell RNA sequencing. Here are the comments on this article:

Major Comments#1:

Do patients with preeclampsia include early-onset and late-onset preeclampsia in the study? Different types of preeclampsia have different pathogenesis and different gestational weeks. The changes in PBMC may be caused by gestational weeks, and the scientific nature of the research remains to be discussed.

Major Comments#2:

The authors used a machine learning method to look for cell type-specific models to predict preeclampsia, but it is inappropriate to call them "predictive" indicators. On the one hand, these indicators are only biomarkers that are characteristically expressed in PE patients. On the other hand, since this study is a cross-sectional study and not a prospective study, these cellular characteristics can be called diagnostic indicators rather than predictive indicators.

Major Comments#3:

The author used all the data for model construction and identified four clusters that could be grouped into disease and normal populations. It is meaningless to use training data for model verification and evaluation. If the author wants to verify the validity of the model, an additional verification, model extrapolation verification, is needed. The method of partial data for training and partial data for verification can be selected. Or they can increase the sample size for verification.

From the above, it is recommended to withdraw the article and review.

Reviewer #2 (Remarks to the Author):

The manuscript aims to characterize the immune cell profile in preeclampsia, to try to predict early diagnosis using machine learning. It is the first paper to include machine learning and RNA profile in immune cells on preeclampsia. Remarkably, the authors give a biological explanation to machine learning results, giving the pathological approach which could be used as a biological marker with further investigation. However, the title only represents some of the work done in the papers since it would be essential to include machine learning in the title as a suggestion. This could be attractive for other investigators to replicate the methodology in another cell line or for further examination in the same field.

According to the work methodology, it is necessary to include the gestational weeks the patients were diagnosed and when the sample was taken. This could improve the understanding of the evolution of the pathology (preeclampsia). Therefore, even when it is described that the sample was at the time of diagnosis in the last paragraph, it is recommended to put it in the Clinical Characteristics section.

Minor comments

In the text, the references to Figure 2 need to be corrected.

Line 128: Text refers to 2D but is 2C

Line 129: Text refers to 2C but is 2D

Line 131: Text refers to 2D but is 2E

Line 134: Text refers to 2E but is 2F
Line 139: Text refers to 2F but is 2G
Line 142: Text refers to 2G but is 2H
Line 149: Text refers to 2H but is 2I

Line 172: It is recommended change 'expansion' to 'increasing'

Figure 4D: In subsection 4D, is it possible to re-order as described in the text?

Figure 4E: It is possible to change subsection E since you previously described the response to type I interferon and MHC Class II.

Line 228: MHC-I, is it referred to as MCH Class II?

Figure 4F: Change VACN to 'VCAN.' Can you re-order as described in the text?

Section Develop of machine-learning: The training data set (Random Forest) and the SHAP method must be included in the text since this is part of the results.

Line 321-324: Please include the idea of 5-fold cross-validation.

Line 324: Why is Figure 6H not included?

Line 329: Is it possible to redraft the idea?

Line 417-419: How are S100 Family genes play a role in preeclampsia?

Line 420-422: Can you discuss more the use of machine learning in this manuscript?

Point-by-Point Response to Reviewer

Reviewer #1 (Remarks to the Author):

Thank you for inviting me to review the article -- Dissecting circulating immune cells of preeclampsia by single-cell RNA sequencing. Here are the comments on this article:

We thank the reviewer for the comments regarding our manuscript. We have modified it according to all his/her suggestions.

Major Comments#1:

Do patients with preeclampsia include early-onset and late-onset preeclampsia in the study? Different types of preeclampsia have different pathogenesis and different gestational weeks. The changes in PBMC may be caused by gestational weeks, and the scientific nature of the research remains to be discussed.

Response #1: We apologize for the incomplete description of our samples. This study included one case of early-onset PE and seven cases of late-onset PE. We agree that gestational ages and other factors can induce changes in PBMC, so we utilized propensity score matching (PSM) to match the control (NP) and balance the confounding factors between the two groups as much as possible. Additionally, we conducted PCA and covariate analysis, which further confirmed that gestational age is not the primary factor contributing to our results. We made the following efforts:

(1) More detailed description: Maternal peripheral blood samples were collected when PE was diagnosed. 15 healthy pregnant women were matched to eliminate the influence of maternal age, gestational age, baby sex, etc. We added the corresponding descriptions in Lines 97-103. Moreover, we have relocated the original clinical information table to the main body of the article as "**Table 1**". In "**New Supplemental Table 1**", we have added the clinical information of each individual in both the NP and PE groups. In "**New Figure 1A**", we have included a timeline depicting the sample collection time points.

New Figure 1A Timeline of sampling

(2) Considering the potential bias in PSM, we conducted both covariate analysis and compared the distribution of principal components before and after gestational age regression, demonstrating that gestational age is not the primary factor causing the change in PBMCs between the two groups.

In the previous version, there were a total of 8 subgroups with statistically significant differences in the proportion of cells in PE and NP groups. The gestational age was used as a covariate to perform a covariate analysis of variance on these subgroups with significant differences:

Response Table 1. Covariate variance analysis of subpopulation cell proportions

Cell type	F value with gestational ages as covariate	P value with gestational ages as covariate	F value between NP and PE after correction of covariate gestational ages	P value between NP and PE after correction of covariate gestational ages
CD8+ NK-like T	0.594	0.4499	6.968	0.0157*
CD160+ NK	0.011	0.91817	9.369	0.00617**
Treg	0.355	0.55819	8.536	0.00844**
XCL1+ NK	0.241	0.6292	5.838	0.0254*
pDC	10.612	0.00394**	4.501	0.04656*
Classical Mono	5.085	0.03549*	10.26	0.00446**
IFN- Non-classical Mono	1.403	0.25013	8.495	0.00857**
Mast	0.383	0.543	5.387	0.031*

The results showed that the proportions of 'CD8⁺ NK-like T', 'CD160⁺ NK', 'Treg' and 'XCL1⁺ NK' in total T & NK cells and 'IFN⁻ Non-classical Mono' and 'Mast' in myeloid cells were not correlated with gestational ages, while the proportions of 'pDC' and 'Classical Mono' were correlated with gestational ages. However, after correcting the influence of the covariate gestational ages, the proportion of cells in all 8 subsets between NP and PE still had significant statistical differences (**Response Table 1**).

Subsequently, we added gestational ages to Metadata, used 'vars.to.regress' in 'SCTransform' function to regress the effect of gestational ages, and performed PCA linear dimension reduction to compare the distribution of PC1 and PC2 before and after regression (**Response Figure 1**). No obvious change indicated that gestational ages was not the main factor affecting the significant change of principal components between the two groups.

Response Figure 1. Distribution of principal components before (left) and after (right) regression

(3) The currently suggested two-stage pathogenesis of preeclampsia involves the first stage of placental abnormalities and the second stage of abnormal maternal response, including systemic inflammatory response^{1, 2}, implying that immune abnormalities in peripheral blood may be a final stage in the pathogenesis of both early-onset and late-onset PE. Therefore, studying the differences between PBMCs from PE and NP could indeed partially reflect the pathological status of preeclampsia. In addition, in two prospective studies that collected maternal blood before diagnosis, cfRNA signatures can robustly predict PE (the studies did not show difference between early-onset and late-onset PE)^{3, 4}. Moreover, transcripts variation of cfRNA was not obscured by the onset subtype, severity, and delivery age of preeclampsia (Before 20 weeks of gestation, a subset of DEGs can separate preeclampsia (PE) and

normotensive samples despite differences in symptom severity, preeclampsia onset subtype and gestational age at delivery⁴). Therefore, in the PBMC study of scRNA-seq, it may be possible to find key immune cells and genes of preeclampsia pathophysiology as potential predictive markers.

Revision made:

Previous supplementary Table to be the main **Table 1** and **New Supplementary Table 1** include each sample clinical characterize.

Line 79-84: In this study, we recruited 8 pregnant women diagnosed with preeclampsia (PE group, included one early-onset PE and seven late-onset PE) and 15 normal pregnant women (NP group) matched by propensity score matching (PSM) to eliminate the effect of confounding factors (**Table 1 and supplemental Table 1**). Blood samples were taken from individuals once they were diagnosed with preeclampsia. The gestational ages at sampling were all in the third trimester, range from 31⁺² to 40⁺³ weeks (**Fig 1A**). All pregnant women enrolled in the observation were nulliparas.

Major Comments#2:

The authors used a machine learning method to look for cell type-specific models to predict preeclampsia, but it is inappropriate to call them "predictive" indicators. On the one hand, these indicators are only biomarkers that are characteristically expressed in PE patients. On the other hand, since this study is a cross-sectional study and not a prospective study, these cellular characteristics can be called diagnostic indicators rather than predictive indicators.

Response #2: We thank the reviewer for pointing out this issue. In the revised version, we have made replacements for words related to "predictive" with terms such as "diagnostic," "diagnose," "distinguish," and so on.

Revision made:

Lines 34-35: Furthermore, we developed four cell-type-specific machine-learning models to identify potential diagnostic indicators of preeclampsia.

Lines 73-74: Moreover, four cell-type-specific machine-learning models were constructed to distinguish preeclampsia from normal pregnancy, and provides new insights of potential biomarkers for diagnosis at single-cell level.

Line 261: **Development of machine-learning models in the important subsets to diagnose preeclampsia.**

Line 521: it is undesirable to build a model to diagnose PE based upon such dataset.

Line 546: The independent testing dataset was then used to evaluate the performance of the classification model.

Line 550: Mean absolute SHAP values across all patients in the dataset represented the overall importance of a particular gene in the diagnosis of PE by the RF model.

Major Comments#3:

The author used all the data for model construction and identified four clusters that could be grouped into disease and normal populations. It is meaningless to use training data for model verification and evaluation. If the author wants to verify the validity of the model, an additional verification, model extrapolation verification, is needed. The method of partial data for training and partial data for verification can be selected. Or they can increase the sample size for

verification.

Response #3: According to the first recommendation that partial data for training and partial data for verification, we randomly divided the original data into a training set and an independent test set in a 7:3 ratio. We separately evaluated the performance of four models on the training set and the independent test set (New Figure 6B-6C and New Supplementary Figure 5). Furthermore, we visualized the results based on the independent test set (Figure 6E-6H). Detailed descriptions of the establishment and validation process of machine learning models can be found in Lines 267-305. The associated parameters for the models are provided in New Supplementary Table 4-5.

Revision made:

New Figure 6B-6C

Supplemental Figure 5

New Supplementary Figure 5

Lines 267-298:

Thus, we sought to develop cell type-specific random forest (RF)-based classifiers for total Mono (VCAN⁺ Mono, classical Mono, IFN⁻ non-classical Mono and IFN⁺ non-classical Mono), CD4⁺ Tn 2, CD8⁺ Tn 2 and Treg to diagnose preeclampsia based on our scRNA-seq data and the results of the expression analysis above (Fig. 6A). Briefly, for each cell type, we first generated pseudo-cells from single cells per individual after single-cell data processing, where each pseudo-cell was labeled as either positive or negative according to the status of the individual (PE or NP).

The random stratified sampling was then applied on the pseudo-cell dataset to create a training set and an independent test set for ensuring the same ratio of positive and negative samples in the two sets (Table S4). Upon the training set after data normalization and feature selection, each cell-type-specific classifier was trained using the 5-fold cross-validation (CV) scheme (80% of random stratified samples used for training while 20% for internal validation). This procedure was repeated 100 times and the optimal hyperparameter values were determined based on the highest AUROC (area under ROC curve) values (Fig. 6B, Table S5). Due to the imbalanced data in this study (positive samples much less than negative ones), we additionally calculated weight F1 scores and confusion matrix to fairly quantify the classification performance. To do this, we utilized the Youden's J statistic to determine the optimal cutoff threshold for each classifier (Fig. 6B), leading to high values of the weight F1 scores (between 0.843 and 0.932) and confusion matrix results (Fig. S5A and S5B). These results demonstrated an expressive performance of each cell-type-specific classifier for preeclampsia diagnosis during the training phase. The final cell-type-specific classifiers were built by re-training the existing ones with the optimal hyperparameters using the whole training sets.

Next, we used the independent test sets to evaluate the preeclampsia diagnosis capabilities of the four classifiers. To assess the stability of each classifier, we employed a bootstrapping approach by randomly sampling the same size of data from the independent test set with replacement 100 times. As a result, the large AUROC values of 0.986 ± 0.011 (mean \pm SD) in total Mono, 0.893 ± 0.040 in CD4⁺ Tn 2, 0.988 ± 0.007 in CD8⁺ Tn 2, and 0.867 ± 0.051 in Treg were reached, respectively (Fig. 6C). Using the optimal cutoff threshold, we additionally calculated the sensitivity (SEN, mean value ranging from 0.821 to 0.936), specificity (SPE, from 0.828 to 1.000), negative predictive value (NPV, from 0.927 to 0.981), and positive predictive value (PPV, from 0.587 to 1.000) of these four models (Fig. 6C), as well as the weighted F1 scores (from 0.839 to 0.942) and confusion matrix results for these four classifiers (Fig. S5C and S5D). Collectively, these results suggest that our models are well-performing to distinguish preeclampsia in both training set and independent test set.

Reviewer #2 (Remarks to the Author):

The manuscript aims to characterize the immune cell profile in preeclampsia, to try to predict early diagnosis using machine learning. It is the first paper to include machine learning and RNA profile in immune cells on preeclampsia. Remarkably, the authors give a biological explanation to machine learning results, giving the pathological approach which could be used as a biological marker with further investigation.

We thank the reviewer for the positive comments regarding our manuscript. We have modified it according to all his/her suggestions.

However, the title only represents some of the work done in the papers since it would be essential to include machine learning in the title as a suggestion. This could be attractive for other investigators to replicate the methodology in another cell line or for further examination in the same field.

Response #1:

Thanks for the suggestion, we included the work of machine learning in the updated title: **Characterize immune variation and diagnostic indicators of preeclampsia by single-cell RNA**

sequencing and machine learning

According to the work methodology, it is necessary to include the gestational weeks the patients were diagnosed and when the sample was taken. This could improve the understanding of the evolution of the pathology (preeclampsia). Therefore, even when it is described that the sample was at the time of diagnosis in the last paragraph, it is recommended to put it in the Clinical Characteristics section.

Response #2: Thank you for the suggestion. Following your recommendation, we have included the gestational ages description in the Clinical Characteristics section. Additionally, we have made extra efforts to provide clearer information about the samples. Firstly, we have added the collection timeline for each sample in **New Figure 1A** (blood samples were collected at the time of diagnosis). The control group (NP) was matched using propensity score matching (PSM) with the disease group (PE), aiming to minimize the impact of gestational ages, maternal age, and newborn gender on the results. Secondly, we have included the clinical information of each sample in **New Supplementary Table 1**. Furthermore, in Response #1 for reviewer #1, we have conducted covariate analysis and compared the top 2 principal components before and after gestational ages regression to demonstrate that gestational ages did not significantly influence our results.

Revision made:

Revision made:

Previous supplementary Table to be the main **Table 1** and **New Supplementary Table 1** include each sample clinical characterize.

Line 79-84: In this study, we recruited 8 pregnant women diagnosed with preeclampsia (PE group, included one early-onset PE and seven late-onset PE) and 15 normal pregnant women (NP group) matched by propensity score matching (PSM) to eliminate the effect of confounding factors (**Table 1 and supplemental Table 1**). Blood samples were taken from individuals once they were diagnosed with preeclampsia. The gestational ages at sampling were all in the third trimester, range from 31⁺² to 40⁺³ weeks (**Fig 1A**). All pregnant women enrolled in the observation were nulliparas.

Minor comments

In the text, the references to Figure 2 need to be corrected.

Line 128: Text refers to 2D but is 2C

Line 129: Text refers to 2C but is 2D

Line 131: Text refers to 2D but is 2E

Line 134: Text refers to 2E but is 2F

Line 139: Text refers to 2F but is 2G

Line 142: Text refers to 2G but is 2H

Line 149: Text refers to 2H but is 2I

Response #3: Thank you for carefully reading our article, and we apologize for any mistakes we have made. We have made the necessary corrections in the references to Figure 2 in Lines 110-130.

Line 172: It is recommended change 'expansion' to 'increasing'

Response #4: Thanks for the suggestion, we have changed 'expansion' to 'increasing'.

Revision made:

Line 153: We observed the **increasing** Treg in PE

Figure 4D: In subsection 4D, is it possible to re-order as described in the text?

Response #5:

Thanks for the suggestion. We re-order Figure 4D as described in the text.

Revision made:

New Figure 4D

Figure 4E: It is possible to change subsection E since you previously described the response to type I interferon and MHC Class II.

Response #6: Thank you for the suggestion. We have added a **new panel of Figure 4E** showing the expression of the 'complement' pathway in the myeloid subsets. We have also observed the upregulation of this pathway in several monocyte subsets. Since both monocytes and B cells are important antigen-presenting cells, we have retained the panel depicting the upregulation of the MHC-II-mediated pathway in monocytes (and moved the previously described IFN-I response to the Supplementary Figure 4B). These two pathways play important roles in the activation of monocytes and inflammatory responses.

Revision made:

Lines 199-206: **In line with the above-mentioned pathways analysis results of lymphocyte subpopulations**, the downregulation of MHC-I and type I IFN response in several myeloid cell subsets **were also identified in PE (Fig. S4B)**. Both B cells and monocytes are important APCs. **Consistent with the result in switched memory B 1**, the MHC-II antigen presentation pathway was also upregulated in three monocyte subsets (VCAN⁺ Mono, classical Mono and IFN⁻ non-classical Mono) **(Fig. 4E)**. **In addition, the complement pathway is also upregulated in monocyte subsets, further indicating that monocyte-mediated immune responses were activated and contribute to the inflammatory response to preeclampsia⁵ (Figure 4E).**

New Figure 4E

Line 228: MHC-I, is it referred to as MCH Class II?

Response #7: We apologize for the unclear description in this section. In the new version, we have adjusted the IFN-I response and MHC-I together in **New Supplementary 4B** and made modifications to provide a clearer textual description in the text.

Revision made:

Lines 199-201: **In line with the above-mentioned pathways analysis results of lymphocyte subpopulations**, the downregulation of MHC-I and type I IFN response in several myeloid cell subsets **were also identified in PE (Fig. S4B)**.

New Supplemental Figure 4B

Figure 4F: Change VACN to 'VCAN.' Can you re-order as described in the text?

Response #8:

We apologize for the mistake in our previous version and have changed 'VACN' to 'VCAN' in the **new Figure 4F**. And we re-order the Figure 4F and Supplementary Figure 4C as described in the text.

Section Develop of machine-learning: The training data set (Random Forest) and the SHAP method must be included in the text since this is part of the results.

Response #9:

We thank the reviewer for pointing out this issue. In the new revision, we have added the missing information for the training data set in Lines 328-347, SHAP method in Lines 365-369; and we have also included the relevant results in the New Figure 6B, Supplemental Figure 5A-5B, and Supplemental Table 4-5.

Revision made:

Lines 267-282: Thus, we sought to develop cell type-specific random forest (RF)-based classifiers for total Mono (VCAN⁺ Mono, classical Mono, IFN⁺ non-classical Mono and IFN⁺ non-classical Mono), CD4⁺ Tn 2, CD8⁺ Tn 2 and Treg to diagnose preeclampsia based on our scRNA-seq data and the results of the expression analysis above (Fig. 6A). Briefly, for each cell type, we first generated pseudo-cells from single cells per individual after single-cell data processing, where each pseudo-cell was labeled as either positive or negative according to the status of the individual (PE or NP). The random stratified sampling was then applied on the pseudo-cell dataset to create a training set and an independent test set for ensuring the same ratio of positive and negative samples in the two sets (Table S4). Upon the training set after data normalization and feature selection, each cell-type-specific classifier was trained using the 5-fold cross-validation (CV) scheme (80% of random stratified samples used for training while 20% for internal validation). This procedure was repeated 100 times and the optimal hyperparameter values were determined based on the highest AUROC (area under ROC curve) values (Fig. 6B, Table S5). Due to the imbalanced data in this study (positive samples much less than negative ones), we additionally calculated weight F1 scores and confusion matrix to fairly quantify the classification performance. To do this, we utilized the Youden's J statistic to determine the optimal cutoff threshold for each classifier (Fig. 6B), leading to high values of the weight F1 scores (between 0.843 and 0.932) and confusion matrix results (Fig. S5A and S5B). These results demonstrated an expressive performance of each cell-type-specific classifier for preeclampsia diagnosis during the training phase. The final cell-type-specific classifiers were built by re-training the existing ones with the optimal hyperparameters using the whole training sets.

Lines 302-306: We prioritized and ranked the gene features in each of the four sets according to

the mean absolute SHapley Additive exPlanation (SHAP⁶) values that were computed across all samples in the corresponding independent test set. A SHAP value (also called feature importance) represented the contribution of a gene feature towards distinguishing PE from NP.

New Figure 6B-6C

Supplemental Figure 5

New Supplemental Figure 5

Line 321-324: Please include the idea of 5-fold cross-validation.

Response #10: Thank you for the suggestion. We have added a description of the 5-fold cross-validation process.

Revision made:

Lines 275-279: Upon the training set after data normalization and feature selection, each cell-type-specific classifier was trained using the 5-fold cross-validation (CV) scheme (80% of random stratified samples used for training while 20% for internal validation). This procedure was repeated 100 times and the optimal hyperparameter values were determined based on the highest AUROC (area under ROC curve) values (Fig. 6B, Table S5).

Line 324: Why is Figure 6H not included?

Response #11: We apologize for the oversight of Figure 6H, and it has been added to the revised submission.

Revision made:

Line 309: After normalization, we noted that the 23 samples were accurately divided into NP

and PE groups (Fig. 6E-H).

Line 329: Is it possible to redraft the idea?

Response #12: Thank you for the suggestion. We have redrafted the idea to make the description clearer.

Revision made:

Lines 313-316: Furthermore, in the classifier of Treg, we observed a decreased expression level of *SPON2*, *NFKBIA*, and the MHC-I molecule *HLA-C* in PE (Fig.6E), which were in line with the downregulation of immune response in PE. *SPON2* is essential for its function as an integrin ligand for the recruitment activities of inflammatory cells⁷.

Line 417-419: How are S100 Family genes play a role in preeclampsia?

Response #12: In our study, we observed upregulation of gene expression of *S100A6*, *S100A8*, and *S100A10* in myeloid cells of PE, which were reported to increase at protein or mRNA levels in PE. A review article has summarized the potential roles of these genes in the pathogenesis of preeclampsia:

The upregulation of *S100A6* was associated with inflammation, which is a characteristic feature of preeclampsia. Additionally, certain proteins, like insulin growth factor binding protein 1 (IGFBP-1), have been found to interact with *S100A6* in a calcium-dependent manner, further implicating its role in preeclampsia⁸. *S100A8*, known as calprotectin, plays a role in immune response, cell proliferation, and apoptosis. It can trigger inflammation and interact with other molecules⁸. *S100A10* appears to be involved in placental differentiation and the proper functioning of mature microvilli⁸.

Since our study focused on peripheral blood samples collected during the third trimester, and most previous research primarily examined placental or serum samples with limited investigation on peripheral blood mononuclear cell (PBMC) subsets, our discussion only pertains to the role of S100 family genes in promoting inflammation in preeclampsia.

Revision made:

Lines 381-382: Finally, the pro-inflammatory S100 family genes were found to be upregulated these monocyte subsets, which could play roles in preeclampsia by contributing to inflammation or interacting with other factors in preeclampsia pregnancy⁸

Line 420-422: Can you discuss more the use of machine learning in this manuscript?

Response #13: Thank you for the suggestion. We have discussed more on machine learning applications in new submission:

Lines 383-395: A handful of studies has built machine-learning models to classify or diagnose disease states or phenotypes using scRNA-seq data^{9, 10, 11}. Here, we utilized circulating immune cells in pregnancy to construct cell-type-specific models for the diagnosis of preeclampsia. Apart from the accurate and effective discrimination of preeclampsia from normal pregnancy at the single-cell subpopulation level, these models also identified the specific genes that may contribute to the disease's pathological processes. Although the lack of scRNA-seq data from PBMCs of preeclampsia prevented us from further validating the performance of our models in external datasets, the successful try of developing the specific models that accounted for cell types heterogeneity may enhance the accuracy of preeclampsia diagnosis. We also provide a framework for analyzing disease characteristics. We could identify disease-related transcriptional changes at one time and construct cell-type-specific models for key subsets of

interest, which may be used as potential biomarkers for diagnosis or therapeutic targets to promote understanding of disease pathophysiology and clinical applications.

Lines 404-410: Finally, the limited sample size involved in the current study may introduce bias to our models for preeclampsia diagnosis, and therefore, further investigation was need to validate the performance using additional scRNA-seq data. However, the results of AUROC, weight F1 scores and other evaluation metrics obtained from the training sets and independent test sets showed the reliability of our classification models, which can also be demonstrated by the identified preeclampsia-associated gene features that were consistent with previous studies.

1. Chappell LC, Cluver CA, Kingdom J, Tong S. Pre-eclampsia. *Lancet* **398**, 341–354 (2021).
2. Laresgoiti-Servitje E, Gomez-Lopez N, Olson DM. An immunological insight into the origins of pre-eclampsia. *Hum Reprod Update* **16**, 510–524 (2010).
3. Rasmussen M, *et al.* RNA profiles reveal signatures of future health and disease in pregnancy. *Nature* **601**, 422–427 (2022).
4. Moufarrej MN, *et al.* Early prediction of preeclampsia in pregnancy with cell-free RNA. *Nature* **602**, 689–694 (2022).
5. Salmon JE, *et al.* Mutations in complement regulatory proteins predispose to preeclampsia: a genetic analysis of the PROMISSE cohort. *PLoS Med* **8**, e1001013 (2011).
6. Lundberg SM, *et al.* From Local Explanations to Global Understanding with Explainable AI for Trees. *Nat Mach Intell* **2**, 56–67 (2020).
7. Jia W, Li H, He YW. The extracellular matrix protein mindin serves as an integrin ligand and is critical for inflammatory cell recruitment. *Blood* **106**, 3854–3859 (2005).
8. Jurewicz E, Filipek A. Ca(2+)- binding proteins of the S100 family in preeclampsia. *Placenta* **127**, 43–51 (2022).
9. Ma Y, *et al.* Accurate Machine Learning Model to Diagnose Chronic Autoimmune Diseases Utilizing Information From B Cells and Monocytes. *Front Immunol* **13**, 870531 (2022).
10. Sehanobish A, Ravindra N, van Dijk D. Gaining insight into SARS-CoV-2 infection and COVID-19 severity using self-supervised edge features and Graph Neural Networks. In: *Proceedings of the AAAI Conference on Artificial Intelligence* (ed[^](eds) (2021).
11. Chen D, *et al.* Single-cell atlas of peripheral blood mononuclear cells from pregnant women. *Clin Transl Med* **12**, e821 (2022).

REVIEWERS' COMMENTS:

Reviewer #2 (Remarks to the Author):

Thanks for the consideration of the observation. It improves the manuscript. Congratulations on the work done.

Point-by-Point Response to Reviewer

REVIEWERS' COMMENTS:

Reviewer #2 (Remarks to the Author):

Thanks for the consideration of the observation. It improves the manuscript. Congratulations on the work done.

Response: We greatly appreciate the reviewer's valuable contributions in improving our work during the review process.